# Negative Feedback Really Matters: Signed Dual-Channel Graph Contrastive Learning Framework for Recommendation

**Leqi Zheng [1], Chaokun Wang [1]\*, Zixin Song [1], Cheng Wu [1],**
**Shannan Yan [1], Jiajun Zhang [2], Ziyang Liu [1]**
[1]Tsinghua University [2]University of Science and Technology of China
{zhenglq24, songzx24, wuc22, ysn24, liu-zy21}@mails.tsinghua.edu.cn
chaokun@tsinghua.edu.cn, zhangjiajun519@mail.ustc.edu.cn

## Abstract

Traditional recommender systems have relied heavily on positive feedback for learning user preferences, while the abundance of negative feedback in real-world scenarios remains underutilized. To address this limitation, recent years have witnessed increasing attention on leveraging negative feedback in recommender systems to enhance recommendation performance. However, existing methods face three major challenges: limited model compatibility, ineffective information exchange, and computational inefficiency. To overcome these challenges, we propose a model-agnostic Signed Dual-Channel Graph Contrastive Learning (SDCGCL) framework that can be seamlessly integrated with existing graph contrastive learning methods. The framework features three key components: (1) a Dual-Channel Graph Embedding that separately processes positive and negative graphs, (2) a Cross-Channel Distribution Calibration mechanism to maintain structural consistency, and (3) an Adaptive Prediction Strategy that effectively combines signals from both channels. Building upon this framework, we further propose a Dual-channel Feedback Fusion (DualFuse) model and develop a two-stage optimization strategy to ensure efficient training. Extensive experiments on four public datasets demonstrate that our approach consistently outperforms state-of-the-art baselines by substantial margins while exhibiting minimal computational complexity. Our source code and data are released at `https://github.com/LQgdwind/nips25-sdcgcl`.

## 1 Introduction

Recommender systems have become integral components of modern digital platforms, significantly influencing user engagement and satisfaction across diverse domains such as e-commerce, social media, and content streaming services. While substantial progress has been made in leveraging positive feedback (e.g., likes, high ratings) for recommendation, the effective utilization of negative feedback (e.g., dislikes, low ratings) remains a critical yet underexplored avenue for enhancing recommendation performance [21, 34, 7, 6].

This disparity is particularly noteworthy given that negative feedback often provides explicit signals about users' preferences and can potentially offer more precise guidance for recommendation refinement than positive feedback alone [22, 52, 43]. As shown in Figure 1, traditional unsigned graphs only capture the existence of positive interactions, whereas sign-aware graphs preserve the polarity of feedback through different edge types, enabling more comprehensive modeling

---

*Corresponding author

39th Conference on Neural Information Processing Systems (NeurIPS 2025).

of user preferences. Therefore, the recent research [34, 29, 6, 43] has increasingly focused on negative feedback in recommender systems. However, these studies face three major challenges: (1) **Limited Model Compatibility**: Most existing methods design specialized models for processing signed feedback [49, 43], making them incompatible with recent advances in graph-based recommendation models. This specialization prevents them from benefiting from state-of-the-art techniques like graph contrastive learning, which have demonstrated remarkable success in unsigned recommendation scenarios. (2) **Limited Information Exchange**: Most existing methods treat negative feedback as auxiliary signals, failing to fully exploit its potential [34, 6]. These methods primarily focus on positive feedback while only partially utilizing negative feedback, resulting in an incomplete understanding of user preferences and suboptimal recommendations. (3) **Limited Training Strategy**: Existing methods either process the full signed graph during training or rely solely on sampled feedback [49, 6], failing to strike a balance between comprehensive learning and training efficiency.

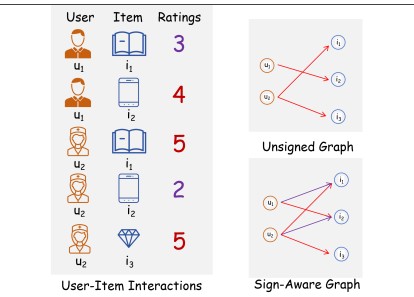

Figure 1: Comparison between unsigned graph and sign-aware graph in recommender systems. Left: User-item interactions with explicit ratings showing both positive (4-5) and negative (1-3) feedback. Right: Unlike unsigned graphs (top), sign-aware graphs (bottom) preserve feedback polarity through different edge types.

Motivated by the aforementioned issues, we propose a novel Signed Dual-Channel Graph Contrastive Learning (SDCGCL) framework that revolutionizes the integration of negative feedback in recommendation systems. The framework consists of three key components: (1) a Dual-Channel Graph Embedding that separately processes positive and negative graphs, (2) a Cross-Channel Distribution Calibration mechanism to maintain structural consistency between channels, and (3) an Adaptive Prediction Strategy that effectively combines signals from both channels. To further enhance the framework's effectiveness, we present the Dual-channel Feedback Fusion (DualFuse) model, which implements a dual-channel graph encoder and cross-channel graph fusion, enabling simultaneous processing of positive and negative feedback patterns. To address training efficiency, we also propose a two-stage optimization strategy that combines comprehensive learning on full graphs with efficient training on strategically sampled subgraphs. This approach is theoretically proven to preserve recommendation quality.

The main contributions of this work are summarized as follows:

- **Model-Agnostic Framework:** We propose SDCGCL, a model-agnostic framework that can be seamlessly incorporated into existing graph contrastive learning methods, overcoming the compatibility limitation of current signed recommendation approaches (Section 2.1).

- **Cross-Channel Information Fusion:** We design DualFuse, a novel model featuring dual-channel encoding and cross-channel fusion mechanisms to enable effective information exchange between positive and negative feedback patterns while preserving channel-specific characteristics (Section 2.2).

- **Two-Stage Training Strategy:** We develop a two-stage optimization strategy combining comprehensive learning on full graphs with efficient training on strategically sampled subgraphs, with theoretical guarantees for both training effectiveness and efficiency (Section 3).

- **Experimental Validation:** Extensive experiments on four public datasets demonstrate that our approach consistently outperforms the state-of-the-art baselines by substantial margins, while achieving the superior computational efficiency with faster convergence (Section 4).

## 2 Model Design

In this section, we propose two key novel techniques: (1) a model-agnostic **Signed Dual-Channel Graph Contrastive Learning** (SDCGCL) framework (Section 2.1), and (2) a **Dual-Channel Feedback Fusion** (DualFuse) model specifically designed to complement this framework (Section 2.2).

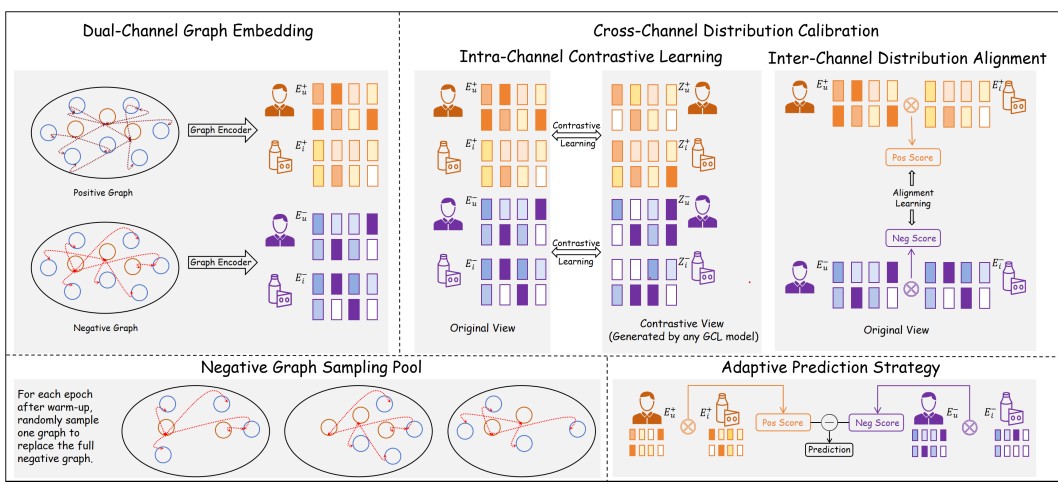

Figure 2: Overview of the SDCGCL framework. The framework consists of three main components: (1) Dual-Channel Graph Embedding, (2) Cross-Channel Distribution Calibration, and (3) Adaptive Prediction Strategy. A negative graph sampling pool (bottom left) enables efficient training optimization.

## 2.1 Model-agnostic SDCGCL Framework

In this subsection, we propose our model-agnostic **Signed Dual-Channel Graph Contrastive Learning** (SDCGCL) framework, which effectively leverages both positive and negative user feedback for recommendation tasks. The SDCGCL framework consists of three key components: dual-channel graph embedding, cross-channel distribution calibration, and an adaptive prediction strategy, as illustrated in Figure 2.

### 2.1.1 Dual-Channel Graph Embedding

To effectively leverage both positive and negative feedback, our SDCGCL framework independently propagates the positive and negative interaction graphs using graph contrastive learning techniques.

For each channel, we independently encode the user and item embeddings through a message propagation function $f(\cdot)$ and a dropout strategy function $p(\cdot)$, which can be instantiated with any suitable GNN backbone. The embeddings are updated over $L$ layers:

$$\mathbf{e}_{u,l}^+ = f\left(p(\hat{\mathbf{A}}^+), \mathbf{e}_{i,l-1}^+\right), \quad \mathbf{e}_{u,l}^- = f\left(p(\hat{\mathbf{A}}^-), \mathbf{e}_{i,l-1}^-\right) \tag{1}$$

where $\mathbf{e}_{u,l}^+$, $\mathbf{e}_{u,l}^-$ denote layer $l$ embeddings, $\mathbf{e}_{i,l-1}^+$, $\mathbf{e}_{i,l-1}^-$ represent layer $l-1$ item embeddings, and $\hat{\mathbf{A}}^+$, $\hat{\mathbf{A}}^-$ are normalized adjacency matrices in positive/negative channels. After $L$ layers of propagation, we obtain the final graph embeddings:

$$\mathbf{E}_u^+ = \mathrm{AGG}\left(\{\mathbf{e}_{u,l}^+ : l \leq L\}\right), \quad \mathbf{E}_u^- = \mathrm{AGG}\left(\{\mathbf{e}_{u,l}^- : l \leq L\}\right) \tag{2}$$

where $\mathbf{E}_u^+$ and $\mathbf{E}_u^-$ denote final graph embeddings in positive/negative channels, and $\mathrm{AGG}(\cdot)$ denotes a function that aggregates embeddings from different layers, such as mean pooling or concatenation.

To generate contrastive views for contrastive learning, we apply the base model's augmentation mechanism to obtain augmented embeddings at a designated layer $l^*$:

$$\mathbf{z}_{u,l^*}^+ = \phi\left(\hat{\mathbf{A}}^+, \mathbf{e}_{u,l}^+, \mathbf{X}, \boldsymbol{\theta}\right), \quad \mathbf{z}_{u,l^*}^- = \phi\left(\hat{\mathbf{A}}^-, \mathbf{e}_{u,l}^-, \mathbf{X}, \boldsymbol{\theta}\right) \tag{3}$$

where $\phi(\cdot)$ represents the base model's specific augmentation mechanism, $\mathbf{X}$ denotes optional node features, and $\boldsymbol{\theta}$ contains augmentation-specific parameters (e.g., dropout rates). The final contrastive embeddings are then obtained by aggregating the augmented embeddings:

$$\mathbf{Z}_u^+ = \mathrm{AGG}^*\left(\{\mathbf{z}_{u,l^*}^+\}\right), \quad \mathbf{Z}_u^- = \mathrm{AGG}^*\left(\{\mathbf{z}_{u,l^*}^-\}\right) \tag{4}$$

where $\mathrm{AGG}^*(\cdot)$ is the contrastive view aggregator. Similar notations apply to items with embeddings $\mathbf{E}_i^+$, $\mathbf{E}_i^-$, $\mathbf{Z}_i^+$ and $\mathbf{Z}_i^-$.

### 2.1.2 Cross-Channel Distribution Calibration.

To effectively integrate information from both positive and negative feedback channels, our framework employs a cross-channel distribution calibration mechanism, which achieves the information integration through two components: intra-channel contrastive learning and inter-channel distribution alignment.

**Intra-Channel Contrastive Learning.** Given that user preferences exhibit inherent structure within both positive and negative interactions, we first employ channel-specific contrastive learning to capture these underlying patterns. For the positive channel, we optimize:

$$\min_{(u,i)\in\mathcal{G}^+} -\{f^*(\mathbf{E}_u^+, \mathbf{Z}_u^+) + f^*(\mathbf{E}_i^+, \mathbf{Z}_i^+) - \sum_{\substack{(u',i')\in\mathcal{G}^+ \\ u'\neq u, i'\neq i}}(\frac{f^*(\mathbf{E}_u^+, \mathbf{Z}_{u'}^+)}{||\mathcal{U}||-1} + \frac{f^*(\mathbf{E}_i^+, \mathbf{Z}_{i'}^+)}{||\mathcal{I}||-1})\} \tag{5}$$

Here, the first two terms $f^*(\mathbf{E}_u^+, \mathbf{Z}_u^+)$ and $f^*(\mathbf{E}_i^+, \mathbf{Z}_i^+)$ maximize agreement between original embeddings and their augmented views, encouraging robustness to perturbations. Similarly, for the negative channel:

$$\min_{(u,i)\in\mathcal{G}^-} -\{f^*(\mathbf{E}_u^-, \mathbf{Z}_u^-) + f^*(\mathbf{E}_i^-, \mathbf{Z}_i^-) - \sum_{\substack{(u',i')\in\mathcal{G}^- \\ u'\neq u, i'\neq i}}(\frac{f^*(\mathbf{E}_u^-, \mathbf{Z}_{u'}^-)}{||\mathcal{U}||-1} + \frac{f^*(\mathbf{E}_i^-, \mathbf{Z}_{i'}^-)}{||\mathcal{I}||-1})\} \tag{6}$$

**Inter-Channel Distribution Alignment** While maintaining channel-specific information is important, the excessive divergence between positive and negative embedding spaces can hinder effective integration. We propose an inter-channel distribution alignment mechanism that enforces structural consistency while preserving distinctive features:

$$\min\{\sum_{u\in\mathcal{U}} g^*(\sum_{(u,i)\in\mathcal{G}^+} \frac{\mathbf{E}_u^+ \circ \mathbf{E}_i^{+\intercal}}{||\mathcal{N}_u^+||}, \sum_{(u,j)\in\mathcal{G}^-} \frac{\mathbf{E}_u^- \circ \mathbf{E}_j^{-\intercal}}{||\mathcal{N}_u^-||})\}, \min\{\sum_{i\in\mathcal{I}} g^*(\sum_{(u,i)\in\mathcal{G}^+} \frac{\mathbf{E}_u^+ \circ \mathbf{E}_i^{+\intercal}}{||\mathcal{N}_i^+||}, \sum_{(v,i)\in\mathcal{G}^-} \frac{\mathbf{E}_v^- \circ \mathbf{E}_i^{-\intercal}}{||\mathcal{N}_i^-||})\} \tag{7}$$

where $g^*(\cdot,\cdot)$ represents a distribution difference measure between positive and negative channels. The first equation aligns user-centric patterns, while the second addresses item-centric alignments, with normalized interaction scores reflecting neighborhood aggregations.

### 2.1.3 Adaptive Prediction Strategy

After obtaining the calibrated embeddings from both channels, we adopt an adaptive prediction strategy to combine them for final recommendation. The predicted preference score $\hat{y}_{u,i}$ for user $u$ and item $i$ is computed by balancing the contributions from the positive and negative embeddings:

$$\hat{y}_{u,i} = (1+k)\mathbf{E}_u^+ \circ \mathbf{E}_i^{+\intercal} - k\mathbf{E}_u^- \circ \mathbf{E}_i^{-\intercal}, \tag{8}$$

where $k \in [0,1]$ is a hyperparameter controlling the influence of negative feedback.

### 2.1.4 Theoretical Analysis

**Theorem 1** (Distribution Instability). *For each node, the negative neighbors $\mathcal{N}^-$ have an unstable degree of scale. For users, some only give positive ratings and refrain from commenting on items they dislike, while others are more direct and express their negative ratings openly. Therefore, in signed recommendation graphs, the negative feedback distribution exhibits higher variance than positive feedback, with embedding distributions satisfying:*

$$\mathbb{E}[\mathbf{E}_u^+ \circ \mathbf{E}_i^{+\intercal}] = \mu, \ \mathbb{E}[\mathbf{E}_u^- \circ \mathbf{E}_i^{-\intercal}] = \delta_1 + \mu$$
$$Var[\mathbf{E}_u^+ \circ \mathbf{E}_i^{+\intercal}] = \sigma^2, \ Var[\mathbf{E}_u^- \circ \mathbf{E}_i^{-\intercal}] = \delta_2\sigma^2 \tag{9}$$

*where $\delta_2 \geq 1$ represents the inherent instability of negative feedback.*

*Proof Sketch.* We show that without distribution alignment, the prediction expectation contains a bias term $-k\delta_1$ and inflated variance dependent on $\delta_2$. Distribution alignment (Eq. 7) ensures $\delta_1 \to 0, \delta_2 \to 1$, normalizing the prediction expectation to $\mathbb{E}[\hat{y}_{u,i}] = \mu$ and simplifying variance to $(2k^2 + 2k + 1) \cdot \sigma^2$. The complete proof is provided in Appendix A.1. $\qquad\square$

## 2.2 DualFuse Model

SDCGCL is model-agnostic and can integrate with various graph contrastive learning models like SGL, XSimGCL, and LightGCL [46, 57, 2]. However, these models lack negative feedback utilization, limiting their ability to fully exploit the potential of our framework. To address this inadequate utilization, we propose DualFuse, which implements dual-channel graph encoding and cross-channel fusion of positive and negative graphs, enabling simultaneous learning of both interaction patterns to maximize SDCGCL's effectiveness.

### 2.2.1 Dual-Channel Graph Encoder

DualFuse employs a dual-channel graph encoder based on LightGCN, with distinct embedding spaces for each channel. Embeddings evolve through layer-wise message propagation within channels, and the final representations are computed via multi-hop connectivity aggregation:

$$\mathbf{E}_u^+ = \frac{\sum_{l=0}^{L} \sum_{i \in \mathcal{N}_u^+} \frac{\mathbf{e}_{i,l-1}^+}{\sqrt{|\mathcal{N}_u^+| \cdot |\mathcal{N}_i^+|}}}{L+1}, \mathbf{E}_u^- = \frac{\sum_{l=0}^{L} \sum_{i \in \mathcal{N}_u^-} \frac{\mathbf{e}_{i,l-1}^-}{\sqrt{|\mathcal{N}_u^-| \cdot |\mathcal{N}_i^-|}}}{L+1} \tag{10}$$

where $\mathbf{E}_u^+$ and $\mathbf{E}_u^-$ denote final graph embeddings in two channels. Similarly, for item $i$, we can obtain $\mathbf{E}_i^+$ and $\mathbf{E}_i^-$.

### 2.2.2 Cross-Channel Graph Fusion

DualFuse leverages an innovative cross-channel graph fusion mechanism where embeddings from each channel create perturbations for the other, enriching representations while preserving channel-specific patterns, as shown in Figure 3.

At a designated layer $l^*$, we generate contrastive views by introducing structured perturbations derived from the opposite channel. This contrastive views implements the data augmentation function $\phi(\cdot)$ from Equation 4 through cross-channel fusion:

$$\mathbf{Z}_u^+ = \frac{1}{L+1} \sum_{l=0}^{L} (\mathbf{e}_{u,l^*}^+ + \frac{\mathbf{e}_{u,l^*}^-}{||\mathbf{e}_{u,l^*}^-||}),$$

$$\mathbf{Z}_u^- = \frac{1}{L+1} \sum_{l=0}^{L} (\mathbf{e}_{u,l^*}^- + \frac{\mathbf{e}_{u,l^*}^+}{||\mathbf{e}_{u,l^*}^+||}) \tag{11}$$

Figure 3: Graph fusion mechanism: Original embeddings from positive (orange) and negative (purple) channels are normalized and fused to generate contrastive views.

where $\mathbf{Z}_u^+$ and $\mathbf{Z}_u^-$ denote final contrastive embeddings in two channels. Similarly, for item $i$, we can obtain $\mathbf{Z}_i^+$ and $\mathbf{Z}_i^-$.

### 2.2.3 Theoretical Analysis

**Theorem 2** (Cross-Channel Information Preservation). *The cross-channel fusion mechanism preserves essential information while maintaining stable gradient flow. For any node $v \in \mathcal{U} \cup \mathcal{I}$, at convergence:*

$$\|\nabla_{\mathbf{e}_v^+} \mathcal{L}\| \approx \|\nabla_{\mathbf{e}_v^-} \mathcal{L}\| \tag{12}$$

*Proof Sketch.* By analyzing gradient propagation through the fusion mechanism, we establish that at convergence, when $\|\frac{\partial \mathcal{L}}{\partial \mathbf{z}_v^+}\| \approx \|\frac{\partial \mathcal{L}}{\partial \mathbf{z}_v^-}\|$ and $\|\mathbf{e}_v^+\| \approx \|\mathbf{e}_v^-\|$, the gradients in both channels maintain similar magnitudes, ensuring balanced information flow. The detailed derivation is available in Appendix A.2. $\qquad\square$

# 3 Optimization Design

To effectively train our framework while managing computational complexity, we propose a two-stage optimization strategy that combines comprehensive learning on the full graph with efficient training on strategically sampled subgraphs.

## 3.1 Two-Stage Optimization Strategy.

**Full-Graph Learning Stage** The first stage operates on the complete signed user-item interaction graph $\mathcal{G}$, processing all positive edges $\mathcal{E}^+$ and negative edges $\mathcal{E}^-$ simultaneously. The full-graph learning stage is crucial for capturing the complete structure of user preferences and ensuring that no valuable negative feedback information is overlooked during the initial training period.

**Sampled-Graph Learning Stage.** To address the computational challenges posed by large-scale negative interaction graphs while maintaining learning effectiveness, we propose a popularity-guided random walk sampling strategy that is formally presented in Algorithm 1. This strategy carefully constructs subgraphs that preserve the most informative negative feedback patterns.

---
**Algorithm 1:** Popularity-Guided Random Walk Sampling

---
**Input** : Original negative graph $\mathcal{G}^-$, sample rate $\rho$, walk length $l$, temperature $\tau$
**Output** : Sampled negative graph $\mathcal{G}^{s-}$
Initialize node degrees $d_v$ for all $v \in \mathcal{V}$;
Compute importance distribution $P(v) \leftarrow \frac{\exp(d_v/\tau)}{\sum_{u \in \mathcal{V}} \exp(d_u/\tau)}$;
$\mathcal{S} \leftarrow$ Sample $\rho|\mathcal{V}|$ nodes according to $P(v)$;
Initialize importance scores $s_v \leftarrow 0$ for all $v \in \mathcal{V}$;
**for** *each starting node* $v_0 \in \mathcal{S}$ **do**
    $v_t \leftarrow v_0$;
    **for** $t = 1$ *to* $l$ **do**
        Compute transition probabilities $P(v_j|v_t) \leftarrow \frac{d_{v_j}}{\sum_{v_k \in \mathcal{N}(v_t)} d_{v_k}}$;
        Sample $v_t$ from $\mathcal{N}(v_t)$ according to $P(v_j|v_t)$;
        Update importance: $s_{v_t} \leftarrow s_{v_t} + d_{v_t} \cdot \frac{l-t}{l}$;
    **end**
**end**
Compute edge importance $w_{ij} \leftarrow \frac{s_i + s_j}{2}$ for each edge $(i, j)$;
Construct $\mathcal{G}^{s-}$ with edges where $w_{ij} > \theta$;
**return** $\mathcal{G}^{s-}$

---

**Theoretical Analysis.** The effectiveness of our two-stage sampling strategy can be theoretically justified through embedding stability bounds:

**Theorem 3** (Two-Stage Stability Bound). *For any node $v \in \mathcal{U} \cup \mathcal{I}$, the expected embedding difference satisfies:*

$$\mathbb{E} \left\| \mathbf{e}_v^{(t)} - \mathbf{e}_v^{(t-1)} \right\|_2^2 \leq \begin{cases} C_1/t, & t \leq T_{warm} \\ C_2(\rho)/t + \epsilon(\rho)/\sqrt{t}, & t > T_{warm} \end{cases} \tag{13}$$

*where $C_1 := \eta_0^2 L^2 D$ integrates the initial learning rate ($\eta_0$), the Lipschitz constant ($L$) of loss gradients in dense interaction regions ($\|\mathbf{e}_u - \mathbf{e}_i\|_2 \geq \delta$), and the maximum node distance ($D$); $C_2(\rho) := \eta_0^2 \left( L^2 + \frac{\sigma_0^2 + \kappa/\rho}{\rho} \right)$ combines gradient smoothness ($L^2$), base variance ($\sigma_0^2$) from full-graph training, and sparse sampling penalty ($\kappa/\rho^2$); $\epsilon(\rho) := \eta_0 \sqrt{\nu(\rho)}$ encodes information loss where $\nu(\rho)$ measures divergence between true and sampled negative feedback distributions.*

*Proof Sketch.* We analyze each optimization stage separately. In the warm-up phase, embedding differences decay as $O(1/t)$. In the sampling phase, the decay follows $O(1/t) + O(1/\sqrt{t})$, reflecting the trade-off between sampling efficiency ($\rho$) and variance control ($\kappa, \nu$). The full derivation is provided in Appendix A.3. $\square$

Table 1: The performance comparison across the methods on four datasets. Baseline best results in **bold**, SDCGCL best results in **bold\***, second-best underlined. Relative improvement (%) shows the performance gain of SDCGCL-DualFuse over the strongest baseline. * indicates statistical significance ($p < 0.01$).

| Group | Datasets | ML-1M | | Yelp | | Amazon | | ML-10M | |
|---|---|---|---|---|---|---|---|---|---|
| | Models | Recall | NDCG | Recall | NDCG | Recall | NDCG | Recall | NDCG |
| U-RS | MF | 0.1329 | 0.1988 | 0.0334 | 0.0217 | 0.0489 | 0.0364 | 0.1667 | 0.2046 |
| | NCF | 0.1501 | 0.2102 | 0.0359 | 0.0237 | 0.0578 | 0.0427 | 0.1926 | 0.2441 |
| | NGCF | 0.1630 | 0.2185 | 0.0566 | 0.0475 | 0.0614 | 0.0463 | 0.2162 | 0.2718 |
| | LightGCN | 0.1993 | 0.2632 | 0.0662 | 0.0539 | 0.0728 | 0.0631 | 0.2597 | 0.3091 |
| | DGCF | 0.1768 | 0.2104 | 0.0629 | 0.0504 | 0.0695 | 0.0613 | 0.2241 | 0.2859 |
| | HyRec | 0.1805 | 0.2181 | 0.0606 | 0.0550 | 0.0658 | 0.0596 | 0.2304 | 0.2805 |
| | GFormer | 0.2272 | 0.2407 | 0.0597 | 0.0542 | 0.0758 | 0.0672 | 0.2460 | 0.3011 |
| | SelfGNN | 0.2565 | 0.2810 | 0.0791 | 0.0672 | 0.0806 | 0.0724 | 0.2742 | 0.3111 |
| | NCL | 0.2627 | 0.2782 | 0.0699 | 0.0615 | 0.0746 | 0.0676 | 0.2972 | 0.3183 |
| | SGL | 0.2798 | 0.3037 | 0.0746 | 0.0729 | 0.0958 | 0.0694 | 0.3056 | 0.3299 |
| | LightGCL | 0.2730 | 0.3035 | 0.0697 | 0.0675 | 0.0967 | 0.0728 | 0.3098 | 0.3231 |
| | XSimGCL | 0.2729 | 0.3087 | 0.0867 | 0.0758 | 0.0963 | 0.0707 | 0.3109 | 0.3371 |
| | IGCL | 0.2747 | 0.3016 | 0.0692 | 0.0660 | 0.0796 | 0.0661 | 0.2956 | 0.3212 |
| S-RS | SiReN | **0.3093** | 0.3338 | 0.0873 | 0.0635 | 0.1017 | 0.0924 | **0.3490** | **0.3583** |
| | SiGRec | 0.1937 | 0.2583 | 0.0594 | 0.0499 | 0.0741 | 0.0678 | 0.2302 | 0.2918 |
| | DFGNN | 0.2538 | 0.3030 | 0.0728 | 0.0609 | 0.0768 | 0.0705 | 0.2721 | 0.3113 |
| | SignGT | 0.1635 | 0.2225 | 0.0607 | 0.0536 | 0.0736 | 0.0644 | 0.2366 | 0.2970 |
| | SBGNN | 0.1527 | 0.2113 | 0.0621 | 0.0479 | 0.0612 | 0.0548 | 0.2237 | 0.2773 |
| | SLGNN | 0.1740 | 0.2370 | 0.0658 | 0.0498 | 0.0617 | 0.0556 | 0.2269 | 0.2912 |
| | SGFormer | 0.1877 | 0.2680 | 0.0601 | 0.0459 | 0.0792 | 0.0648 | 0.2344 | 0.2908 |
| | SIGFormer | 0.2995 | **0.3380** | 0.0856 | 0.0777 | 0.1006 | 0.0997 | 0.3217 | 0.3549 |
| | NFARec | 0.2840 | 0.3212 | **0.0971** | **0.0808** | **0.1136** | **0.1020** | 0.3316 | 0.3442 |
| Ours | SDCGCL-SGL | 0.2879 | 0.3300 | 0.1136 | 0.0907 | 0.1108 | 0.1001 | 0.3768 | 0.3716 |
| | SDCGCL-LightGCL | 0.2945 | 0.3423 | 0.1069 | 0.0826 | 0.1057 | 0.0890 | 0.3810 | 0.3760 |
| | SDCGCL-XSimGCL | 0.3050 | 0.3401 | 0.1112 | 0.0881 | 0.1142 | 0.1014 | 0.3791 | 0.3726 |
| | SDCGCL-DualFuse | **0.3282\*** | **0.3693\*** | **0.1243\*** | **0.0959\*** | **0.1342\*** | **0.1113\*** | **0.3900\*** | **0.3860\*** |
| Relative improvement (%) | | 6.110% | 9.260% | 28.012% | 18.689% | 18.133% | 9.118% | 11.748% | 7.731% |

## 3.2 Multi-Objective Loss Integration

Our training approach integrates three key loss components to effectively capture both positive and negative feedback patterns: (1) recommendation loss based on Bayesian Personalized Ranking for supervision, (2) contrastive learning loss to enhance embedding quality within each channel, and (3) distribution alignment loss to maintain consistent structural information between channels. The final objective function combines these components with balanced weighting parameters:

$$\mathcal{L} = \mathcal{L}_{rec} + \lambda(\mathcal{L}_{cl} + \gamma\mathcal{L}_{dist}) + \eta||\Theta||_2^2 \tag{14}$$

where $\lambda$ controls the overall contribution of the auxiliary objectives, $\gamma$ weights the distribution alignment constraint, and $\eta$ is the $L_2$ regularization coefficient applied to model parameters $\Theta$. A comprehensive description of each loss component and their mathematical formulations is provided in Appendix B.

# 4 Experiments

## 4.1 Experimental Setup

We evaluate our method on four publicly available recommendation datasets: Yelp, Amazon, and MovieLens (ML-1M and ML-10M). Following established conventions, we binarize ratings (scores $\geq 4$ as positive, $< 4$ as negative). For the performance evaluation, we adopt Recall@20 and NDCG@20 as metrics. We benchmark against 22 state-of-the-art recommendation methods across unsigned and sign-aware recommendation systems. Detailed descriptions of datasets, metrics, baselines, and parameter settings are provided in Appendix C.

## 4.2 Overall Performance

Experimental evaluations demonstrate our approach's superior performance. As shown in Table 1, DualFuse consistently outperforms all baselines by substantial margins. Compared to the best

unsigned baseline (XSimGCL), our model achieves 20.26% and 19.63% improvements in Recall@20 and NDCG@20 on ML-1M. Against sign-aware methods, DualFuse shows significant gains over NFARec, with Recall@20 improvements of 6.11% (ML-1M), 28.01% (Yelp), and 18.13% (Amazon). SDCGCL framework enhances all integrated methods (see Appendix D.2), validating our dual-channel architecture's effectiveness.

## 4.3 Ablation Study

### 4.3.1 Component Analysis

We evaluate our framework through ablation studies on key components. Table 2 shows removing fusion causes 73.80% Recall@20 decrease, while CL and alignment provide 10.45% and 3.49% NDCG@20 improve-

Table 2: Performance analysis with different component combinations

| Variant | Components | | | | Performance | |
|---------|-----------|-----|-------|-----|--------|------|
| | Fusion | CL | Align | Rec | Recall | NDCG |
| DualFuse | ✓ | ✓ | ✓ | ✓ | **0.3282** | **0.3693** |
| w/o Fusion | ✗ | ✓ | ✓ | ✓ | 0.0860 | 0.0857 |
| w/o CL | ✓ | ✗ | ✓ | ✓ | 0.2983 | 0.3307 |
| w/o Align | ✓ | ✓ | ✗ | ✓ | 0.3231 | 0.3564 |
| w/o Rec | ✓ | ✓ | ✓ | ✗ | 0.2436 | 0.2586 |

ments respectively. Recommendation loss is critical, with its removal causing 69.94% NDCG@20 degradation and scattered embedding distributions (Details see Appendix D.3 and Figure 4).

### 4.3.2 Impact of Sampling Rate

Our sampling rate analysis empirically validates the theoretical bounds established in Theorem 3 through three critical regimes of operation. The full-graph training setting ($\rho = 1.0$) corre-

Table 3: Analysis of sampling rate optimization

| $\rho$ | Performance | | Efficiency | | |
|--------|-------------|------|------------|-------|-------|
| | Recall | NDCG | Time/epoch | Conv. | Total |
| 1.0 | 0.3226 | 0.3567 | 75.30s | 23 | 28.9m |
| 0.1 | 0.3268 | 0.3604 | 68.03s | 21 | 23.8m |
| 0.01 | **0.3282** | **0.3693** | 59.57s | 21 | 20.8m |
| 0.001 | 0.2025 | 0.2050 | 50.01s | >50 | >41.7m |

sponds to the $C_1/t$-dominated warm-up phase in our theoretical framework. While this configuration achieves reasonable performance (0.3226 Recall@20), it requires 28.9 minutes total training time, demonstrating the computational cost of unoptimized stability bounds. This setting serves as our baseline for comparison with sampling-optimized approaches.

The optimal sampling configuration ($\rho = 0.01$) represents a critical balance point between the competing terms in our theoretical model. This rate effectively balances the $C_2(\rho)/t$ term (where $\kappa/\rho^2$ is bounded at $10^4 \times \kappa$) and the $\epsilon(\rho)/\sqrt{t}$ term from Theorem 3. The experimental results confirm that this configuration delivers the peak performance (0.3282 Recall@20) while requiring only 20.8 minutes of training, a 28.0% efficiency gain compared to full-graph training. This empirical finding aligns with the $O(1/t)$ decay advantage predicted by our theoretical analysis.

## 4.4 Empirical Analysis

**Robustness Analysis** To evaluate the model robustness, we conduct experiments by randomly corrupting 0%-20% of user-item interactions. SDCGCL-DualFuse demonstrates the superior stability, maintaining 81.3% of its performance under 20% noise on MovieLens and 76.7% on Amazon. As shown in Figure 5, all SDCGCL variants exhibit improved robustness compared to their base counterparts, attributed to our dual-channel architecture and cross-channel calibration.

**Parameter Sensitivity** Our framework involves five key hyperparameters that control different aspects of model behavior. Through extensive analysis, we observe that moderate values consistently yield optimal performance: channel balancing parameter $\alpha$ (0.1-0.4), contrastive learning parameter $\beta$ (0.3-0.7), distribution alignment parameter $\gamma$ (0.1-0.5), and negative feedback weight $k$ (0.1-0.3). The auxiliary loss parameter $\lambda$ shows stability in the range of 0.1-0.2. Figure 6 visualizes these effects across datasets. These findings confirm our theoretical analysis in Section 2.1.4, particularly regarding the necessity of distribution alignment.

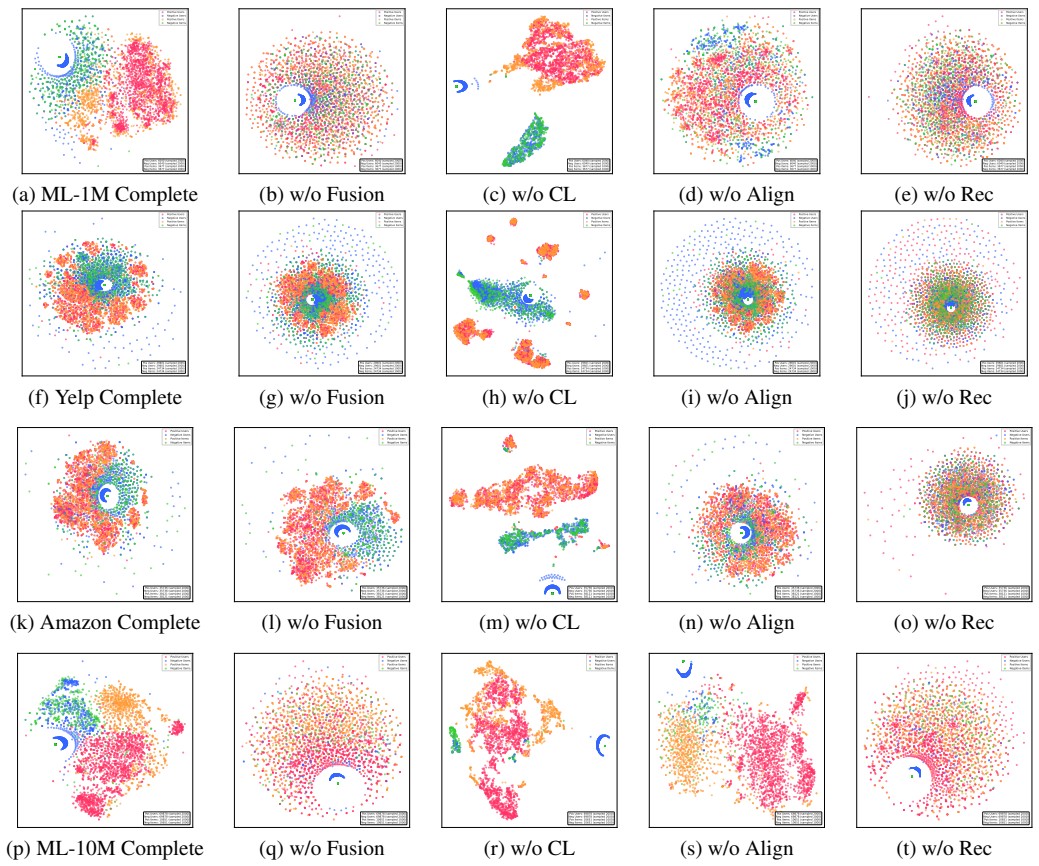

Figure 4: t-SNE visualization of learned embeddings on four datasets across different ablation settings. Red/orange: positive users/items; blue/green: negative users/items.

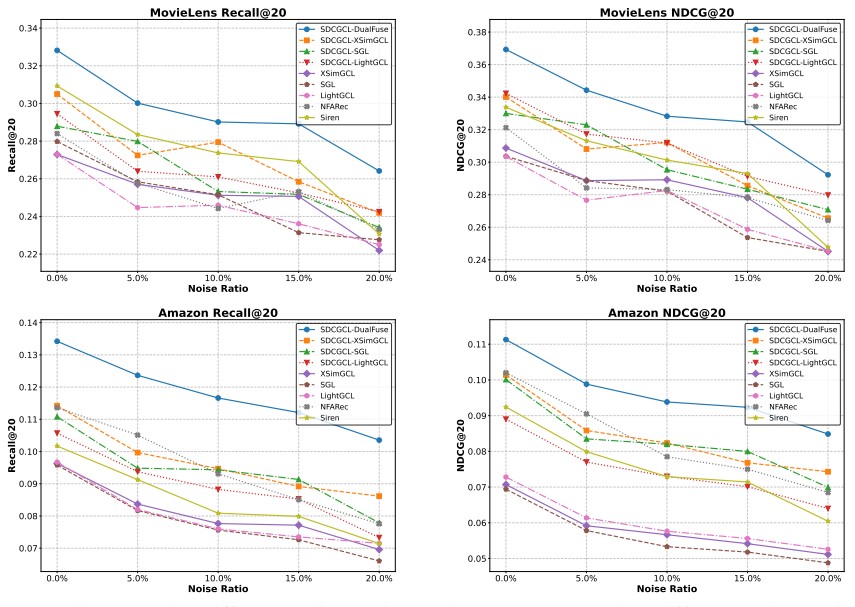

(a) Recall@20 under Different Noise Ratios    (b) NDCG@20 under Different Noise Ratios

Figure 5: Model robustness evaluation on two datasets showing superior performance stability of SDCGCL variants under increasing noise ratios.

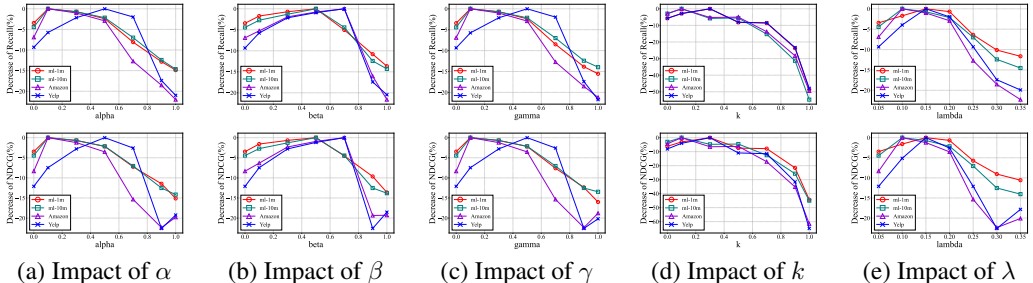

| (a) Impact of $\alpha$ | (b) Impact of $\beta$ | (c) Impact of $\gamma$ | (d) Impact of $k$ | (e) Impact of $\lambda$ |

Figure 6: Parameter sensitivity analysis showing impact on Recall@20 and NDCG@20 across four datasets.

## 4.5 More Detailed Experiments

We provide additional experimental results in the appendix, including efficiency analysis (Appendix D.1), performance improvement analysis (Appendix D.2), and extended ablation studies (Appendix D.3) to supplement the main findings.

## 5 Related Work

### 5.1 GNNs for Recommendation

Deep learning[59, 37, 11, 26], especially GNN, has revolutionized recommender systems through their capacity to model complex user-item relationships. Starting with foundational works [33, 35], the field evolved through message passing innovations like GC-MC [36] and NGCF [40], reaching a significant milestone with LightGCN [16] and GCCF [5]. Subsequent developments enhanced theoretical foundations through pre-training [14], filtering mechanisms [58], and multi-view learning [61], while temporal modeling advanced through architectures like SRGNN [48], GCE-GNN [44], and TGSREC [10]. Dynamic patterns were captured by DGCF [41], TG-MC [1], DGSR [60], and SURGE [3], while contrastive learning emerged as a promising direction [2, 27, 57, 55, 53, 65, 64], further enhanced by transformer integrations [42, 51, 50]. However, these methods struggle with heterogeneous feedback types.

### 5.2 Sign-aware Recommendation

Sign-aware recommendation systems evolved from foundational explicit feedback methods like user-based CF [63], PMF [30], and SVD++ [23]. Built upon theoretical foundations in spectral analysis [18], matrix decomposition [19], and balance theory [17, 7], contemporary research has deepened understanding of negative feedback [22, 52], leading to innovations in graph-based systems [29, 34], sampling strategies [8, 9], interactive platforms [62], and sequential models [31, 32]. While various approaches have interpreted different user behaviors as negative signals [39, 12, 45, 62], many methods still struggle with effective integration, either excluding negative instances [54, 56] or oversimplifying interactions. Our work addresses these limitations through a model-agnostic framework that seamlessly integrates with existing methods while avoiding traditional balance theory constraints [34].

## 6 Conclusion

In this paper, we propose SDCGCL, a novel model-agnostic framework for effectively leveraging negative feedback in recommender systems, along with DualFuse, a specially designed model that maximizes the framework's capabilities. Through theoretical analysis and extensive experiments, we demonstrate how our dual-channel architecture, cross-channel distribution calibration mechanism, and adaptive prediction strategy successfully address the fundamental challenges of incorporating negative feedback while maintaining computational efficiency. The framework's model-agnostic nature enables seamless integration with existing graph contrastive learning methods, consistently yielding substantial performance improvements across multiple datasets and baseline models. Our comprehensive empirical results validate that negative feedback indeed plays a crucial role in enhancing recommendation performance when properly utilized through our proposed framework.

## Acknowledgments

This work is supported in part by the National Natural Science Foundation of China (No. 62372264 and No. 92467203) and Sina Weibo Corp. Chaokun Wang is the corresponding author.

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

## A   Detailed Proofs

### A.1   Complete Proof of Theorem 2.1: Distribution Instability

*Proof.* We begin by establishing the distributional properties of embeddings derived from positive and negative feedback channels. Given the assumptions stated in the theorem, we proceed as follows:

Let us denote $X_{ui}^+ = \mathbf{E}_u^+ \circ \mathbf{E}_i^{+\top}$ and $X_{ui}^- = \mathbf{E}_u^- \circ \mathbf{E}_i^{-\top}$ as the interaction scores in positive and negative channels, respectively. By assumption, these random variables follow distributions with

$$
\begin{aligned}
\mathbb{E}[X_{ui}^+] = \mu, && \text{Var}[X_{ui}^+] = \sigma^2 \\
\mathbb{E}[X_{ui}^-] = \mu + \delta_1, && \text{Var}[X_{ui}^-] = \delta_2 \sigma^2
\end{aligned}
\tag{15}
$$

where $\delta_1$ represents the mean shift and $\delta_2 \geq 1$ captures the increased variance in negative feedback distributions.

For the predicted preference score $\hat{y}_{u,i} = (1+k)X_{ui}^+ - kX_{ui}^-$, we derive its expectation:

$$
\begin{aligned}
\mathbb{E}[\hat{y}_{u,i}] &= \mathbb{E}[(1+k)X_{ui}^+ - kX_{ui}^-] \\
&= (1+k)\mathbb{E}[X_{ui}^+] - k\mathbb{E}[X_{ui}^-] \\
&= (1+k)\mu - k(\mu + \delta_1) \\
&= (1+k)\mu - k\mu - k\delta_1 \\
&= \mu - k\delta_1
\end{aligned}
\tag{16}
$$

Assuming independence between channels, we derive the variance:

$$
\begin{aligned}
\text{Var}[\hat{y}_{u,i}] &= \text{Var}[(1+k)X_{ui}^+ - kX_{ui}^-] \\
&= (1+k)^2 \text{Var}[X_{ui}^+] + k^2 \text{Var}[X_{ui}^-] \\
&= (1+k)^2 \sigma^2 + k^2 \delta_2 \sigma^2 \\
&= \sigma^2[(1+k)^2 + k^2 \delta_2] \\
&= \sigma^2[1 + 2k + k^2 + k^2 \delta_2] \\
&= \sigma^2[1 + 2k + k^2(1 + \delta_2)]
\end{aligned}
\tag{17}
$$

This demonstrates that without distribution alignment, the prediction has a systematic bias of $-k\delta_1$ and inflated variance scaled by $\delta_2 \geq 1$.

When applying our cross-channel distribution calibration mechanism (Equation 7), we enforce $\delta_1 \to 0$ and $\delta_2 \to 1$. Consequently:

$$
\begin{aligned}
\mathbb{E}[\hat{y}_{u,i}] &= \mu \\
\text{Var}[\hat{y}_{u,i}] &= \sigma^2[1 + 2k + k^2(1 + 1)] \\
&= \sigma^2(1 + 2k + 2k^2)
\end{aligned}
\tag{18}
$$

Thus, the alignment mechanism eliminates the bias term $-k\delta_1$ from the prediction expectation and normalizes the variance to a more stable form that depends solely on $k$ rather than the instability parameter $\delta_2$, completing the proof. $\square$

### A.2 Complete Proof of Theorem 2.2: Cross-Channel Information Preservation

*Proof.* We analyze the gradient flow through the cross-channel fusion mechanism to demonstrate that it maintains balanced information flow between positive and negative channels.

1. **Gradient Analysis for Positive Channel:**

For the positive channel fusion operation, the gradient with respect to the embedding $\mathbf{e}_v^+$ can be expressed as:

$$
\begin{aligned}
\nabla_{\mathbf{e}_v^+} \mathcal{L} &= \frac{\partial \mathcal{L}}{\partial \mathbf{e}_v^+} \\
&= \frac{\partial \mathcal{L}}{\partial \mathbf{Z}_v^+} \frac{\partial \mathbf{Z}_v^+}{\partial \mathbf{e}_v^+} + \frac{\partial \mathcal{L}}{\partial \mathbf{Z}_v^-} \frac{\partial \mathbf{Z}_v^-}{\partial \mathbf{e}_v^+}
\end{aligned}
\tag{19}
$$

The first term corresponds to the direct gradient flow within the positive channel, while the second term captures the cross-channel influence through the fusion mechanism.

From Equation 13 in the main paper, we have:

$$\mathbf{Z}_v^+ = \frac{1}{L+1} \sum_{l=0}^{L} \left( \mathbf{e}_{v,l^*}^+ + \frac{\mathbf{e}_{v,l^*}^-}{\|\mathbf{e}_{v,l^*}^-\|} \right)$$

$$\mathbf{Z}_v^- = \frac{1}{L+1} \sum_{l=0}^{L} \left( \mathbf{e}_{v,l^*}^- + \frac{\mathbf{e}_{v,l^*}^+}{\|\mathbf{e}_{v,l^*}^+\|} \right)$$

(20)

Therefore:

$$\frac{\partial \mathbf{Z}_v^+}{\partial \mathbf{e}_v^+} = \frac{1}{L+1}$$

$$\frac{\partial \mathbf{Z}_v^-}{\partial \mathbf{e}_v^+} = \frac{1}{L+1} \frac{\partial}{\partial \mathbf{e}_v^+} \left( \frac{\mathbf{e}_v^+}{\|\mathbf{e}_v^+\|} \right)$$

(21)

The derivative of the normalized vector can be expanded as:

$$\frac{\partial}{\partial \mathbf{e}_v^+} \left( \frac{\mathbf{e}_v^+}{\|\mathbf{e}_v^+\|} \right) = \frac{\partial}{\partial \mathbf{e}_v^+} \left( \frac{\mathbf{e}_v^+}{\sqrt{\mathbf{e}_v^+ \cdot \mathbf{e}_v^+}} \right)$$

$$= \frac{1}{\|\mathbf{e}_v^+\|} \mathbf{I} - \frac{\mathbf{e}_v^+ \mathbf{e}_v^{+T}}{\|\mathbf{e}_v^+\|^3}$$

(22)

where $\mathbf{I}$ is the identity matrix.

For simplicity and to understand the upper bound, we can establish:

$$\left\| \frac{\partial}{\partial \mathbf{e}_v^+} \left( \frac{\mathbf{e}_v^+}{\|\mathbf{e}_v^+\|} \right) \right\| \leq \frac{1}{\|\mathbf{e}_v^+\|}$$

(23)

## 2. **Gradient Norm Bounds:**

Using the above results, we can now bound the gradient norm:

$$\|\nabla_{\mathbf{e}_v^+} \mathcal{L}\| = \left\| \frac{\partial \mathcal{L}}{\partial \mathbf{Z}_v^+} \frac{\partial \mathbf{Z}_v^+}{\partial \mathbf{e}_v^+} + \frac{\partial \mathcal{L}}{\partial \mathbf{Z}_v^-} \frac{\partial \mathbf{Z}_v^-}{\partial \mathbf{e}_v^+} \right\|$$

$$\leq \left\| \frac{\partial \mathcal{L}}{\partial \mathbf{Z}_v^+} \frac{\partial \mathbf{Z}_v^+}{\partial \mathbf{e}_v^+} \right\| + \left\| \frac{\partial \mathcal{L}}{\partial \mathbf{Z}_v^-} \frac{\partial \mathbf{Z}_v^-}{\partial \mathbf{e}_v^+} \right\|$$

$$\leq \left\| \frac{\partial \mathcal{L}}{\partial \mathbf{Z}_v^+} \right\| \frac{1}{L+1} + \left\| \frac{\partial \mathcal{L}}{\partial \mathbf{Z}_v^-} \right\| \frac{1}{L+1} \frac{1}{\|\mathbf{e}_v^+\|}$$

$$= \frac{1}{L+1} \left( \left\| \frac{\partial \mathcal{L}}{\partial \mathbf{Z}_v^+} \right\| + \left\| \frac{\partial \mathcal{L}}{\partial \mathbf{Z}_v^-} \right\| \frac{1}{\|\mathbf{e}_v^+\|} \right)$$

(24)

Similarly, for the negative channel:

$$\|\nabla_{\mathbf{e}_v^-} \mathcal{L}\| \leq \frac{1}{L+1} \left( \left\| \frac{\partial \mathcal{L}}{\partial \mathbf{Z}_v^-} \right\| + \left\| \frac{\partial \mathcal{L}}{\partial \mathbf{Z}_v^+} \right\| \frac{1}{\|\mathbf{e}_v^-\|} \right)$$

(25)

## 3. **Establishing Lower Bounds:**

We can also establish lower bounds:

$$\|\nabla_{\mathbf{e}_v^+}\mathcal{L}\| \geq \frac{1}{L+1}\left(\left\|\frac{\partial\mathcal{L}}{\partial\mathbf{Z}_v^+}\right\| - \left\|\frac{\partial\mathcal{L}}{\partial\mathbf{Z}_v^-}\right\|\frac{1}{\|\mathbf{e}_v^+\|}\right)$$

$$\|\nabla_{\mathbf{e}_v^-}\mathcal{L}\| \geq \frac{1}{L+1}\left(\left\|\frac{\partial\mathcal{L}}{\partial\mathbf{Z}_v^-}\right\| - \left\|\frac{\partial\mathcal{L}}{\partial\mathbf{Z}_v^+}\right\|\frac{1}{\|\mathbf{e}_v^-\|}\right) \tag{26}$$

## 4. Convergence Analysis:

At convergence, several key conditions are satisfied:

1) The loss gradients with respect to contrastive embeddings from both channels become approximately equal: $\left\|\frac{\partial\mathcal{L}}{\partial\mathbf{Z}_v^+}\right\| \approx \left\|\frac{\partial\mathcal{L}}{\partial\mathbf{Z}_v^-}\right\|$.

2) The embedding norms from both channels converge to similar magnitudes: $\|\mathbf{e}_v^+\| \approx \|\mathbf{e}_v^-\|$.

Substituting these conditions into our bounds:

$$\frac{1}{L+1}\left(1 - \frac{1}{\|\mathbf{e}_v^+\|}\right)\left\|\frac{\partial\mathcal{L}}{\partial\mathbf{Z}_v^+}\right\| \leq \|\nabla_{\mathbf{e}_v^+}\mathcal{L}\| \leq \frac{1}{L+1}\left(1 + \frac{1}{\|\mathbf{e}_v^+\|}\right)\left\|\frac{\partial\mathcal{L}}{\partial\mathbf{Z}_v^+}\right\|$$

$$\frac{1}{L+1}\left(1 - \frac{1}{\|\mathbf{e}_v^-\|}\right)\left\|\frac{\partial\mathcal{L}}{\partial\mathbf{Z}_v^-}\right\| \leq \|\nabla_{\mathbf{e}_v^-}\mathcal{L}\| \leq \frac{1}{L+1}\left(1 + \frac{1}{\|\mathbf{e}_v^-\|}\right)\left\|\frac{\partial\mathcal{L}}{\partial\mathbf{Z}_v^-}\right\| \tag{27}$$

Since $\left\|\frac{\partial\mathcal{L}}{\partial\mathbf{Z}_v^+}\right\| \approx \left\|\frac{\partial\mathcal{L}}{\partial\mathbf{Z}_v^-}\right\|$ and $\|\mathbf{e}_v^+\| \approx \|\mathbf{e}_v^-\|$ at convergence, we can conclude that:

$$\|\nabla_{\mathbf{e}_v^+}\mathcal{L}\| \approx \|\nabla_{\mathbf{e}_v^-}\mathcal{L}\| \tag{28}$$

This demonstrates that our cross-channel fusion mechanism maintains balanced gradient flow between positive and negative channels, ensuring that both channels contribute roughly equally to the learning process despite their potentially different initial characteristics. $\square$

### A.3 Complete Proof of Theorem 3.1: Two-Stage Stability Bound

*Proof.* We'll analyze each stage of our optimization process separately.

## 1. Warmup Stage Analysis (Full-Graph Learning):

During the warmup stage ($t \leq T_{warm}$), we train on the complete graph without sampling. Under standard assumptions for gradient-based optimization, the loss function $\mathcal{L}$ has $L$-Lipschitz gradients in dense interaction regions where $\|\mathbf{e}_u - \mathbf{e}_i\|_2 \geq \delta$, the maximum distance between any node embeddings is bounded by $D$ and the learning rate is initialized as $\eta_0$ and potentially follows a schedule.

For gradient descent with these conditions, we have:

$$\mathbf{e}_v^{(t)} - \mathbf{e}_v^{(t-1)} = -\eta_{t-1}\nabla_{\mathbf{e}_v}\mathcal{L}(\Theta^{(t-1)})$$

$$\|\mathbf{e}_v^{(t)} - \mathbf{e}_v^{(t-1)}\|_2^2 = \eta_{t-1}^2\|\nabla_{\mathbf{e}_v}\mathcal{L}(\Theta^{(t-1)})\|_2^2 \tag{29}$$

By the Lipschitz gradient assumption in dense regions:

$$\|\nabla_{\mathbf{e}_v}\mathcal{L}(\Theta^{(t-1)})\|_2^2 \leq L^2 D \tag{30}$$

With learning rate schedule $\eta_t = \eta_0/\sqrt{t+1}$, we get:

$$\begin{aligned}
\mathbb{E}\|\mathbf{e}_v^{(t)} - \mathbf{e}_v^{(t-1)}\|_2^2 &\leq \eta_{t-1}^2 L^2 D \\
&= \frac{\eta_0^2}{t}L^2 D \\
&= \frac{C_1}{t}
\end{aligned} \tag{31}$$

where $C_1 = \eta_0^2 L^2 D$ is our warmup stage constant.

## 2. Sampling Stage Analysis (Subgraph Learning):

After the warmup stage $(t > T_{warm})$, we transition to training on sampled subgraphs. This introduces two additional sources of variance: *Sampling Variance*: Due to sampling a subset of the graph with rate $\rho$ and *Distribution Divergence*: Information loss from potentially missing important negative feedback.

Let's denote the true gradient as $\nabla\mathcal{L}$ and the sampled gradient as $\tilde{\nabla}\mathcal{L}$. The variance of the sampled gradient can be decomposed as:

$$
\begin{aligned}
\mathrm{Var}(\tilde{\nabla}\mathcal{L}) &= \mathbb{E}\|\tilde{\nabla}\mathcal{L} - \nabla\mathcal{L}\|_2^2 + \mathbb{E}\|\nabla\mathcal{L} - \mathbb{E}[\nabla\mathcal{L}]\|_2^2 \\
&= \underbrace{\frac{\sigma_0^2}{\rho}}_{\text{base variance}} + \underbrace{\frac{\kappa}{\rho^2}}_{\text{sparsity penalty}} + \underbrace{\nu(\rho)}_{\text{information loss}} ,
\end{aligned}
\tag{32}
$$

where $\sigma_0^2$ is the base variance from full-graph training , $\kappa$ is a constant that scales the variance inflation from rare negative feedback and $\nu(\rho)$ measures the information loss due to potential systematic bias in sampled graphs.

The embedding difference now satisfies:

$$
\begin{aligned}
\mathbb{E}\|\mathbf{e}_v^{(t)} - \mathbf{e}_v^{(t-1)}\|_2^2 &= \mathbb{E}\| - \eta_{t-1}\tilde{\nabla}_{\mathbf{e}_v}\mathcal{L}(\Theta^{(t-1)})\|_2^2 \\
&= \eta_{t-1}^2 \mathbb{E}\|\tilde{\nabla}_{\mathbf{e}_v}\mathcal{L}(\Theta^{(t-1)})\|_2^2
\end{aligned}
\tag{33}
$$

The expected squared norm of the sampled gradient can be decomposed as:

$$
\begin{aligned}
\mathbb{E}\|\tilde{\nabla}_{\mathbf{e}_v}\mathcal{L}(\Theta^{(t-1)})\|_2^2 &= \|\mathbb{E}[\tilde{\nabla}_{\mathbf{e}_v}\mathcal{L}(\Theta^{(t-1)})]\|_2^2 + \mathrm{Var}(\tilde{\nabla}_{\mathbf{e}_v}\mathcal{L}(\Theta^{(t-1)})) \\
&= \|\nabla_{\mathbf{e}_v}\mathcal{L}(\Theta^{(t-1)}) - \beta(\rho)\|_2^2 + \mathrm{Var}(\tilde{\nabla}_{\mathbf{e}_v}\mathcal{L}(\Theta^{(t-1)}))
\end{aligned}
\tag{34}
$$

where $\beta(\rho)$ represents the bias introduced by sampling, which has magnitude proportional to $\sqrt{\nu(\rho)}$.

Combining these results:

$$
\begin{aligned}
\mathbb{E}\|\mathbf{e}_v^{(t)} - \mathbf{e}_v^{(t-1)}\|_2^2 &\le \eta_{t-1}^2\left(L^2 + \frac{\sigma_0^2}{\rho} + \frac{\kappa}{\rho^2}\right) + \eta_{t-1}\sqrt{\nu(\rho)} \\
&= \frac{\eta_0^2}{t}\left(L^2 + \frac{\sigma_0^2}{\rho} + \frac{\kappa}{\rho^2}\right) + \frac{\eta_0}{\sqrt{t}}\sqrt{\nu(\rho)} \\
&= \frac{C_2(\rho)}{t} + \frac{\epsilon(\rho)}{\sqrt{t}}
\end{aligned}
\tag{35}
$$

where $C_2(\rho) = \eta_0^2(L^2 + \frac{\sigma_0^2 + \kappa/\rho}{\rho})$ and $\epsilon(\rho) = \eta_0\sqrt{\nu(\rho)}$.

This demonstrates that the embedding differences decay as $O(1/t)$ during the warmup phase and as $O(1/t) + O(1/\sqrt{t})$ during the sampling phase. The additional $O(1/\sqrt{t})$ term reflects the trade-off between sampling efficiency (parameter $\rho$) and approximation quality (parameters $\kappa$ and $\nu$). $\qquad\square$

# B   Detailed Multi-Objective Loss Integration

This section provides comprehensive details on our training approach that integrates recommendation supervision, contrastive learning signals, and distribution alignment to effectively capture both types of feedback patterns.

**Algorithm 2:** SDCGCL Optimization Algorithm

---

**Input** : Positive graph $\mathcal{G}^+$, negative graph $\mathcal{G}^-$, total epochs $T$, warm-up epochs $T_{warm}$, hyperparameters $\alpha, \beta, \lambda, \gamma, \eta_r$, learning rate $\eta$
**Output :** Trained model parameters $\Theta$
Initialize model parameters $\Theta$ randomly;

**for** $t = 1$ **to** $T$ **do**
    **if** $t \leq T_{warm}$ **then**
        | $\mathcal{G}_s^- \leftarrow \mathcal{G}^-$ ;                      /* Full negative graph for warm-up */
    **else**
        | $\mathcal{G}_s^- \leftarrow$ PGRWSampling$(\mathcal{G}^-, \rho)$ ;                   /* Algorithm 1 */
    **end**
    **for** *each mini-batch $\mathcal{B}$* **do**
        /* Dual-channel embedding generation                          */
        $\mathbf{E}^+, \mathbf{Z}^+ \leftarrow$ GraphEncoder$(\mathcal{G}^+, \mathcal{B})$;
        $\mathbf{E}^-, \mathbf{Z}^- \leftarrow$ GraphEncoder$(\mathcal{G}_s^-, \mathcal{B})$;
        /* Multi-objective loss computation                      */
        $\mathcal{L}_{rec} \leftarrow (1-\alpha)\mathcal{L}_{rec}^+(\mathbf{E}^+) + \alpha\mathcal{L}_{rec}^-(\mathbf{E}^-)$;
        $\mathcal{L}_{cl} \leftarrow (1-\beta)\mathcal{L}_{cl}^+(\mathbf{E}^+, \mathbf{Z}^+) + \beta\mathcal{L}_{cl}^-(\mathbf{E}^-, \mathbf{Z}^-)$;
        $\mathcal{L}_{dist} \leftarrow$ JS$(\mathbf{E}^+, \mathbf{E}^-)$ ;             /* Jensen-Shannon divergence */
        /* Total loss with regularization                         */
        $\mathcal{L} \leftarrow \mathcal{L}_{rec} + \lambda(\mathcal{L}_{cl} + \gamma\mathcal{L}_{dist}) + \eta_r\|\Theta\|_2^2$;
        /* Parameter update                               */
        $\Theta \leftarrow$ Adam$(\Theta, \nabla_\Theta\mathcal{L}, \eta)$;
    **end**
**end**
**return** $\Theta$

---

## B.1 Recommendation Loss

We adopt Bayesian Personalized Ranking (BPR) loss across both channels. For positive channel:

$$\mathcal{L}_{rec}^+ = -\sum_{(u,i,j)\in\mathcal{O}^+} \ln \sigma(\hat{y}_{u,i} - \hat{y}_{u,j}) \tag{36}$$

where $\mathcal{O}^+$ contains positive triplets $(u,i,j)$ with user $u$, observed item $i$, and unobserved item $j$. Similarly, for the negative channel, we can obtain $\mathcal{L}_{rec}^-$, the overall recommendation loss combines both channels:

$$\mathcal{L}_{rec} = (1-\alpha)\mathcal{L}rec^+ + \alpha\mathcal{L}_{rec}^- \tag{37}$$

## B.2 Contrastive Loss

Following Equations 5 and 6, we apply InfoNCE loss within each channel. For positive channel:

$$\mathcal{L}_{cl}^+ = -\sum_{u\in\mathcal{U}} \ln \frac{\exp(sim(\mathbf{E}_u^+, \mathbf{Z}_u^+)/\tau)}{\sum_{v\in\mathcal{U}}\exp(sim(\mathbf{E}_u^+, \mathbf{Z}_v^+)/\tau)} - \sum_{i\in\mathcal{I}} \ln \frac{\exp(sim(\mathbf{E}_i^+, \mathbf{Z}_i^+)/\tau)}{\sum_{j\in\mathcal{I}}\exp(sim(\mathbf{E}_i^+, \mathbf{Z}_j^+)/\tau)} \tag{38}$$

where $sim(\cdot, \cdot)$ is dot product similarity and $\tau$ controls distribution sharpness. Similarly, for the negative channel, we can obtain $\mathcal{L}_{cl}^-$, the combined loss is:

$$\mathcal{L}_{cl} = (1-\beta)\mathcal{L}_{cl}^+ + \beta\mathcal{L}_{cl}^- \tag{39}$$

## B.3 Distribution Alignment Loss

For Inter-Channel Distribution Alignment (Equations 7), we use Jensen-Shannon divergence:

$$\mathcal{L}_{dist} = \sum_{u\in\mathcal{U}} JS(\sum_{i\in\mathcal{N}_u^+} \frac{\mathbf{E}_u^+ \circ \mathbf{E}_i^{+\mathsf{T}}}{||\mathcal{N}_u^+||}, \sum_{i\in\mathcal{N}_u^-} \frac{\mathbf{E}_u^- \circ \mathbf{E}_i^{-\mathsf{T}}}{||\mathcal{N}_u^-||}) + \sum_{i\in\mathcal{I}} JS(\sum_{u\in\mathcal{N}_i^+} \frac{\mathbf{E}_u^+ \circ \mathbf{E}_i^{+\mathsf{T}}}{||\mathcal{N}_i^+||}, \sum_{u\in\mathcal{N}_i^-} \frac{\mathbf{E}_u^- \circ \mathbf{E}_i^{-\mathsf{T}}}{||\mathcal{N}_i^-||})$$
$$\tag{40}$$

where normalized interaction scores correspond to neighborhood aggregation terms and JS divergence serves as $g^*(\cdot, \cdot)$.

## B.4 Joint Training Objective

The final training objective combines all three components with balanced weighting parameters:

$$\mathcal{L} = \mathcal{L}_{rec} + \lambda(\mathcal{L}_{cl} + \gamma\mathcal{L}_{dist}) + \eta||\Theta||_2^2 \tag{41}$$

where $\lambda$ controls the overall contribution of the auxiliary objectives, $\gamma$ weights the distribution alignment constraint, and $\eta$ is the $L_2$ regularization coefficient applied to model parameters $\Theta$.

During training, we implement a two-stage optimization strategy where the full negative graph $\mathcal{G}^-$ is used for the initial $T_{warm}$ epochs (warm-up stage), followed by our efficient Popularity-Guided Random Walk Sampling approach for subsequent epochs. This warm-up period establishes stable initial representations while the sampling stage significantly reduces computational costs without sacrificing model performance.

Empirically, we found that setting $T_{warm} = 1$ provides sufficient initialization while minimizing overhead, enabling our model to achieve superior results with reduced training time. The sampling rate $\rho$ controls the subgraph size and directly affects the efficiency-quality trade-off as demonstrated in Section 4.3. The complete optimization process is formally presented in Algorithm 2.

## C Detailed Experimental Setup

### C.1 Datasets

We evaluate our method on four publicly available recommendation datasets: Yelp (business reviews), Amazon (book reviews), and MovieLens (movie ratings) with varying scales, as detailed in Table 4. Following established conventions [42, 56, 43], we binarize all numerical ratings by considering scores $\geq 4$ as *positive feedback* and scores $< 4$ as *negative feedback*. Each dataset is rigorously partitioned into training, validation, and test sets using a 7:1:2 ratio to prevent data leakage and ensure reproducible evaluation.

Table 4: Statistics of datasets. #Pos/#Neg refers to the percentage of positive and negative samples.

| Dataset | #Users | #Items | #Interaction | #Pos/#Neg |
|---|---|---|---|---|
| Yelp[1] | 29,601 | 24,734 | 2,074,594 | 66.3%/33.7% |
| Amazon[2] | 35,736 | 38,121 | 1,960,674 | 80.6%/19.4% |
| ML-1M[3] | 6,040 | 3,706 | 1,000,209 | 57.5%/42.5% |
| ML-10M[3] | 69,878 | 10,677 | 10,000,054 | 58.9%/41.1% |

### C.2 Metrics

For performance evaluation, we adopt two standard ranking metrics: Recall@K, which measures the ratio of correctly recommended items over all ground truth items, and NDCG@K, which considers both the hit ratio and position of correctly recommended items. Following previous studies on graph-based recommendation [16, 2, 57, 6], we set K = 20 in our experiments.

### C.3 Baselines

In our experimental evaluation, we benchmark our SDCGCL framework against a diverse set of 22 contemporary recommendation methods. These approaches can be divided into two main categories.

- **Unsigned RS:** Traditional methods like **MF** [23] and **NCF** [15] focus on basic collaborative filtering. Graph-based approaches including **NGCF** [40], **LightGCN** [16], and **DGCF**

---

[1] https://business.yelp.com/data/resources/open-dataset/
[2] https://cseweb.ucsd.edu/~jmcauley/datasets/amazon_v2/
[3] https://grouplens.org/datasets/movielens/

[41] leverage various graph neural architectures. Advanced frameworks such as **HyRec** [38], **GFormer** [24], and **SelfGNN** [28] explore specialized structures. Recent contrastive learning methods (**NCL** [27], **SGL** [46], **LightGCL** [2], **XSimGCL** [57], **IGCL** [13]) enhance representation learning through different augmentation strategies.

- **Sign-aware RS:** Early approaches (**SiReN** [34], **SiGRec** [21], **DFGNN** [49]) focus on separate processing of positive and negative feedback. Transformer-based methods including **SignGT** [4], **SGFormer** [47], and **SIGformer** [6] leverage attention mechanisms. Other specialized architectures like **SBGNN** [20], **SLGNN** [25], and **NFARec** [43] explore unique graph structures and operators for signed feedback.

## C.4 Experiment Setting

For our SDCGCL, we adopt the Adam optimizer and employ grid search for hyperparameter optimization. Specifically, we set the hidden embedding dimension $d$ to 64. The learning rate is set to $10^{-3}$ with a batch size of 2048. The hyperparameter search ranges are $\alpha \in [0, 1]$, $\beta \in [0, 1]$, $\gamma \in [0, 1]$, $k \in [0, 1]$ and $\lambda \in [0.05, 0.35]$. For computational efficiency, $T_{warm}$ is uniformly set to 1. For other baseline models, we strictly follow their officially released code to ensure fair comparison. All experiments in this paper are conducted on 8 RTX3090 GPUs.

# D Additional Experiments

## D.1 Computational Efficiency Analysis

Our SDCGCL framework introduces minimal computational overhead (3-8s per epoch) while significantly improving convergence speed across different base models. SDCGCL-DualFuse achieves the best efficiency with the fastest convergence (21 epochs) and lowest total training time (20.8 minutes) on ML-1M dataset, benefiting from both the enhanced learning signals of negative feedback and efficient architecture design. Compared to the base models (SGL, LightGCL, XSimGCL), our framework consistently reduces the overall training time by improving convergence despite the slight increase in per-epoch processing time.

We provide a detailed analysis of the time complexity for our framework and its comparison with other methods.

Table 5: Runtime comparison across methods on ML-1M dataset.

| Method | Time/epoch | Epochs | Total time |
|---|---|---|---|
| SGL | 81.09s | 32 | 43.25m |
| SDCGCL-SGL | 89.57s | 27 | 40.31m |
| LightGCL | 66.09s | 23 | 25.3m |
| SDCGCL-LightGCL | 69.57s | 24 | 27.83m |
| XSimGCL | **56.09s** | 25 | 23.3m |
| SDCGCL-XSimGCL | 59.34s | 23 | 22.74m |
| SDCGCL-DualFuse | 59.57s | **21** | **20.8m** |

Table 6: Detailed time complexity comparison across methods. $q$: preserved features in LightGCL SVD; $a$: augmentation ratio in SGL; $\rho_0$: the two-stage optimization weighting coefficient; $\mathcal{M} = |\mathcal{U}| + |\mathcal{I}|$: total nodes.

| Method | Augmentation | Convolution | BPR | Contrast | Align | Time/Epoch |
|---|---|---|---|---|---|---|
| SGL | $O(2a|\mathcal{E}^+|)$ | $O(2|\mathcal{E}^+|Ld + 4a|\mathcal{E}^+|Ld)$ | $O(2\mathcal{M}d)$ | $O(\mathcal{M}d)$ | - | 81.09s |
| SDCGCL-SGL | $O(2a|\mathcal{E}^+| + 2a\rho_0|\mathcal{E}^-|)$ | $O((2+4a)(|\mathcal{E}^+| + \rho_0|\mathcal{E}^-|)Ld)$ | $O(2\mathcal{M}d)$ | $O(\mathcal{M}d)$ | $O(\mathcal{M}d)$ | 89.57s |
| LightGCL | - | $O(2|\mathcal{E}^+|Ld + 2q\mathcal{M}Ld)$ | $O(2\mathcal{M}d)$ | $O(\mathcal{M}d)$ | - | 66.09s |
| SDCGCL-LightGCL | - | $O(2(|\mathcal{E}^+| + \rho_0|\mathcal{E}^-|)Ld + 4q\mathcal{M}Ld)$ | $O(2\mathcal{M}d)$ | $O(\mathcal{M}d)$ | $O(\mathcal{M}d)$ | 69.57s |
| XSimGCL | - | $O(2|\mathcal{E}^+|Ld)$ | $O(2\mathcal{M}d)$ | $O(\mathcal{M}d)$ | - | **56.09s** |
| SDCGCL-XSimGCL | - | $O(2|\mathcal{E}^+|Ld + 2\rho_0|\mathcal{E}^-|Ld)$ | $O(2\mathcal{M}d)$ | $O(\mathcal{M}d)$ | $O(\mathcal{M}d)$ | 59.34s |
| SDCGCL-DualFuse | - | $O(2|\mathcal{E}^+|Ld + 2\rho_0|\mathcal{E}^-|Ld)$ | $O(2\mathcal{M}d)$ | $O(\mathcal{M}d)$ | $O(\mathcal{M}d)$ | 59.57s |

As a preprocessing step, SDCGCL employs a negative graph sampling strategy with complexity $O(k\rho\mathcal{M})$, where $k$ denotes the number of sampled neighbors and $\rho$ represents the sampling ratio. We define $\rho_0$ as the two-stage optimization weighting coefficient, whose value is $(epoch - T_{warm})\rho + T_{warm}$. Since $T_{warm} = 1 \ll epoch$, therefore $\rho_0 \approx \rho$.

The additional computational overhead introduced by SDCGCL varies across different base models, with complexity $O(2\rho_0|\mathcal{E}^-|Ld)$ for XSimGCL, $O(2q\mathcal{M}Ld)$ for LightGCL, and $O(2a\rho_0|\mathcal{E}^-|Ld)$ for SGL. DualFuse, specifically designed for efficient implementation of the SDCGCL framework, maintains the same simplified convolution complexity $O(2|\mathcal{E}^+|Ld + 2\rho_0|\mathcal{E}^-|Ld)$ as SDCGCL-XSimGCL while avoiding additional operations like feature decomposition or augmentation required by other variants.

The theoretical analysis of time complexity can only reveal the complexity of each epoch, while the actual running time is also affected by factors such as convergence speed. Our empirical results in Table 5 demonstrate that SDCGCL framework consistently improves training efficiency across different base models by introducing minimal computational overhead while significantly accelerating convergence in most cases.

## D.2  Performance Gains Analysis

Table 7 presents a comprehensive analysis of performance enhancements achieved by the SDCGCL framework when integrated with existing recommendation approaches. The empirical evidence demonstrates consistent and substantial improvements across multiple datasets and baseline architectures.

Table 7: Performance enhancement analysis of the SDCGCL framework relative to baseline models. Results are reported as Recall@20/NDCG@20 pairs, with improvement percentages indicating relative performance gains across metrics.

| Dataset | Base Model | Original | SDCGCL-Enhanced | Improvement (%) |
|---|---|---|---|---|
| ML-1M | SGL | 0.2798/0.3037 | 0.2879/0.3300 | +2.89%/+8.66% |
| | LightGCL | 0.2730/0.3035 | 0.2945/0.3423 | +7.88%/+12.78% |
| | XSimGCL | 0.2729/0.3087 | 0.3050/0.3401 | +11.76%/+10.17% |
| Yelp | SGL | 0.0746/0.0729 | 0.1136/0.0907 | +52.28%/+24.42% |
| | LightGCL | 0.0697/0.0675 | 0.1069/0.0826 | +53.37%/+22.37% |
| | XSimGCL | 0.0867/0.0758 | 0.1112/0.0881 | +28.26%/+16.23% |
| Amazon | SGL | 0.0958/0.0694 | 0.1108/0.1001 | +15.66%/+44.24% |
| | LightGCL | 0.0967/0.0728 | 0.1057/0.0890 | +9.31%/+22.25% |
| | XSimGCL | 0.0963/0.0707 | 0.1142/0.1014 | +18.59%/+43.42% |
| ML-10M | SGL | 0.3056/0.3299 | 0.3768/0.3716 | +23.30%/+12.64% |
| | LightGCL | 0.3098/0.3231 | 0.3810/0.3760 | +22.98%/+16.37% |
| | XSimGCL | 0.3109/0.3371 | 0.3791/0.3726 | +21.94%/+10.53% |

## D.3  Extended Ablation Analysis

We evaluate the contribution of each key component by conducting ablation studies on: (1) cross-channel fusion ("w/o Fusion") from the DualFuse base model, and three components from our SDCGCL framework: (2) contrastive learning ("w/o CL"), (3) distribution alignment ("w/o Align"), and (4) recommendation loss ("w/o Rec"). Figure 4 and Table 8 present the visualization and quantitative results across all datasets. The experimental results reveal that cross-channel fusion

Table 8: Ablation study on different components of SDCGCL. The base model is the DualFuse. Best results are highlighted in **bold**.

| Variant | ML-1M | | Yelp | | Amazon | | ML-10M | |
|---|---|---|---|---|---|---|---|---|
| | Recall | NDCG | Recall | NDCG | Recall | NDCG | Recall | NDCG |
| DualFuse | **0.3282** | **0.3693** | **0.1243** | **0.0959** | **0.1342** | **0.1113** | **0.3900** | **0.3860** |
| w/o Fusion | 0.0860 | 0.0857 | 0.1046 | 0.0894 | 0.1274 | 0.1063 | 0.1624 | 0.1494 |
| w/o CL | 0.2983 | 0.3307 | 0.0969 | 0.0835 | 0.1161 | 0.0950 | 0.3780 | 0.3699 |
| w/o Align | 0.3231 | 0.3564 | 0.1032 | 0.0896 | 0.1297 | 0.1086 | 0.3847 | 0.3751 |
| w/o Rec | 0.2436 | 0.2586 | 0.0793 | 0.0667 | 0.0680 | 0.0530 | 0.1535 | 0.1389 |

exhibits dataset-dependent impact. Its removal causes severe performance degradation on MovieLens (73.80% decrease in Recall@20 on ML-1M, 58.36% decrease on ML-10M) but moderate impact on Yelp (15.85% decrease) and minimal impact on Amazon (5.07% decrease). This suggests that cross-channel fusion is particularly critical for datasets with dense user-item interaction patterns like MovieLens.

The contrastive learning component consistently contributes to model performance across all datasets, with its removal causing 10.45% decrease in NDCG@20 on ML-1M, 12.93% on Yelp, 14.65% on Amazon, and 4.17% on ML-10M. Similarly, distribution alignment shows moderate but consistent impact across datasets, with performance drops of 3.49% on ML-1M, 6.57% on Yelp, 2.43% on Amazon, and 2.82% on ML-10M when removed.

Most importantly, the recommendation loss proves essential for effective training across all datasets, as its removal leads to randomly scattered embedding distributions (Figure 4(e)) and the most substantial performance drops: 69.94% decrease in NDCG@20 on ML-1M, 30.45% on Yelp, 52.38% on Amazon, and 64.02% on ML-10M. This demonstrates the fundamental role of supervised signals in learning discriminative representations for different types of feedback, regardless of dataset characteristics.

## E   Limitations and Future Work

While our proposed SDCGCL framework demonstrates significant improvements across multiple benchmarks, there are several avenues for future exploration. The current implementation has been validated on public benchmark datasets, but deployment in large-scale industrial recommender systems might introduce additional complexities that deserve further investigation. Additionally, the framework could be extended to capture dynamic evolution of user feedback patterns over time, which might further enhance recommendation performance in dynamic environments. Future work could also explore the integration of explicit explanation mechanisms to improve user experience and trust, as well as adaptation strategies for extreme cold-start scenarios where both positive and negative feedback signals are initially limited.

## F   Broader Impacts

Our work focuses on enhancing both the performance and efficiency of recommender systems through effective utilization of negative feedback, thereby benefiting the overall development of recommendation technologies. The proposed framework improves user experience across various digital platforms including e-commerce, social media, and content streaming services. We do not foresee any negative impacts resulting from our work.

