# OpenReview forum: "Negative Feedback Really Matters: Signed Dual-Channel Graph Contrastive Learning Framework for Recommendation"
_NeurIPS.cc/2025/Conference — NeurIPS 2025 poster_

### Official Review · Reviewer_tuXv · 2025-07-01

**Clarity:** 2
**Significance:** 3
**Originality:** 3
**Rating:** 5
**Confidence:** 4

**Summary:**

The paper presents SDCGCL, short for Signed Dual-Channel Graph Contrastive Learning, a novel graph-based approach for recommendation which adopts contrastive learning and works on both positive and negative feedback of users. Specifically, the model builds upon a known limitation in the current literature, where almost no recommendation model leverages negative feedback to exploit the full spectrum of users' preferences over items.

The proposed framework involves the following steps. To begin with, two parallel graph neural networks are used to encode users and items embeddings for positive and negative feedback graphs, where an additional views of the embeddings are obtained in each considered graph at a specific propagation layer through an augmentation strategy on the node embeddings.

Second, a contrastive learning strategy is adopted to capture similarities between the constructed views within the same feedback graph (i.e., positive and negative), named as intra-channel contrastive learning. At the same time, an inter-channel distribution alignment mechanism is implied to avoid an excessive divergence between the learned positive and negative embeddings, as the two graphs (for the different feedback) can largely be largely divergent in their distribution. Finally, the two learned embeddings (positive and negative) are used to predict the user-item interaction with different importance weight $k$, a hyper-parameter of the model.

In addition to the above, the authors specify that the overall training consists of two separate phases. First, they implement a full-graph learning strategy, where both positive and negative graphs are exploited in their integrity to allow the model to learn global perspectives on the whole dataset. Soon after that, the authors propose to perform a random-walk sampling on the negative graph (based on items' popularity) to ease the computational complexity to work on the full negative feedback graph, which is (intuitively) way larger than the positive feedback one. The overall loss function is a multi-objective combination of different losses coming from the previous modules of the framework.

Empirical results on four popular recommendation datasets and against a very large plethora of recommender systems spanning unsigned and signed recommendation demonstrate the superior performance provided by the proposed approach. An ablation study further justifies the architectural choices made by the authors, where the fusion module appears to be the most fundamental one among all the proposed solutions. Additional analyses on the efficiency, robustness, parameter sensitivity, and a case study on distribution alignment, complement the evaluation of the model, which seem to work pretty well under many different perspectives.

**Questions:**

My questions are mainly referred to some of the weaknesses I raised above.

Q1) Regarding the data augmentation function in equation (3), why did the authors decide to apply it to the embeddings instead of, say, to the adjacency matrix. Did they decide to perform an embedding dropout strategy instead of more traditional edge or node dropout? And, if so, why did they do that?

Q2) Regarding the sampled-graph learning stage, could choosing a random-walk strategy based upon items' popularity impact on the diversity of recommendations? In other words, do the authors think that sampling the graphs based upon the items' popularity might enhance the visualization of popular items at the detriment of the niche ones? For instance, I would suggest the authors perform additional evaluation on metrics such as the average percentage of long-tail items (APLT [\*]) for the popularity bias and the expected free discovery (EFD [\*\*]) for recommendation diversity.

**Suggestions**

I would further suggest the authors take a look at this quite novel recommendation dataset [\*\*\*] with incorporates actual negative feedback from users. I think it could be very useful to further analyze the goodness of their methodology.

**References**

[\*] Himan Abdollahpouri, Robin Burke, and Bamshad Mobasher. 2017. Controlling Popularity Bias in Learning-to-Rank Recommendation. In RecSys. ACM, 42–46.

[\*\*] S. Vargas, P. Castells, Rank and relevance in novelty and diversity metrics for recommender systems, in: RecSys, ACM, 2011, pp. 109–116.

[\*\*\*] Chongming Gao, Shijun Li, Wenqiang Lei, Jiawei Chen, Biao Li, Peng Jiang, Xiangnan He, Jiaxin Mao, and Tat-Seng Chua. 2022. KuaiRec: A Fully-observed Dataset and Insights for Evaluating Recommender Systems. In Proceedings of the 31st ACM International Conference on Information &amp; Knowledge Management (CIKM '22). Association for Computing Machinery, New York, NY, USA, 540–550. https://doi.org/10.1145/3511808.3557220

**Ethical Concerns:**

["NO or VERY MINOR ethics concerns only"]

**Final Justification:**

After the rebuttal phase, after reading the other reviews, and authors' responses to those, I decide to confirm my initial score I gave to the paper.

Specifically, all my concerns have been discussed and addressed by the authors.

Whatever the final outcome for this paper, I think it will benefit quite a lot from the fruitful discussion we had along with the authors, the reviewers, and the Area Chair. I believe the paper will improve its quality a lot in any case.

**Limitations:**

Yes.

**Paper Formatting Concerns:**

I do not see any major formatting issues in the paper.

**Quality:**

3

**Strengths And Weaknesses:**

**Strengths**

$\bullet$ The paper tackles an understudied but quite important aspect in recommendation, namely, the usage of explicitly negative feedback. Very few papers in the literature are addressing this topic, and this is also evidenced by the lack of many datasets explicitly incorporating this information.

$\bullet$ The related work is detailed, and helps placing the proposed approach quite well in the current literature, outlining its positive aspects with respect to existing solutions.


$\bullet$ The methodology seems adequately sound, and all proposed solutions are always supported by theoretical proofs (provided in the Supplementary Materials). I particularly appreciated the calibration strategy between negative and positive feedback embeddings, as it is quite clear that the latter would largely diverge over the former without a tailored solution to address this problem. I also really liked the two stage optimization which allows to train the model on the global and local views of the graphs.

$\bullet$ The experimental setting is very extensive, with many baselines spanning signed and unsigned recommender systems, and several evaluation dimensions which help providing a large set of perspectives on the model's performance and architectural choices. Moreover, code is released at review time to ensure the full reproducibility of the work.

**Weaknesses**

$\bullet$ Even acknowledging the page limitations, I think the paper would have benefitted from having a starting (short) section on the background and notations. Indeed, starting off with embedding and graph notations, considering that not all the readers might be fully expert on the topic, could represent a slight challenge for the average reader. I would suggest the authors add a background/preliminaries section at least in the Supplementary Material.

$\bullet$ Regarding the data augmentation function in equation (3), I am wondering why the authors decided to apply it to the embeddings instead of, say, to the adjacency matrix. Did they decide to perform an embedding dropout strategy instead of more traditional edge or node dropout? And, if so, why did they do that? I think this aspect should be better clarified in the methodology.

$\bullet$ Regarding the sampled-graph learning stage, I am wondering if choosing a random-walk strategy based upon items' popularity might impact on the diversity of recommendations. In other words, do the authors think that sampling the graphs based upon the items' popularity might enhance the visualization of popular items at the detriment of the niche ones? I think this aspect should be further clarified by the authors, maybe also by measuring additional recommendation metrics which account for popularity bias and diversity in the recommendation lists.

$\bullet$ Even acknowledging the pages limitations and formatting, I think it would have been beneficial to have Table 1 in the following page with respect to where it is right now, just for the sake of easy readability along with the respective section in the text.

---

> ### Author Rebuttal · Authors · 2025-07-30
>
> **Reviewer Comment:**
>
> > Q1&W2: Regarding the data augmentation function in equation (3), I am wondering why the authors decided to apply it to the embeddings instead of, say, to the adjacency matrix. Did they decide to perform an embedding dropout strategy instead of more traditional edge or node dropout? And, if so, why did they do that? I think this aspect should be better clarified in the methodology.
>
> **Response:**
>
> Thank you for this excellent clarification question. Our φ(·) function in Equation (3) serves as a **general interface** that captures augmented embeddings from different base models, regardless of how the underlying augmentation is performed.
>
> The key design principle is model compatibility. When our framework integrates with SGL, the base model continues to perform its edge dropout on adjacency matrices during message passing, and φ(·) captures the resulting embeddings computed from the modified graph structure. Similarly, when integrated with LightGCL, φ(·) represents embeddings derived from SVD-reconstructed graphs, while with XSimGCL, it captures embeddings after embedding-level perturbations.
>
> This abstraction enables our model-agnostic framework to work seamlessly with different augmentation strategies without forcing base models to change their preferred approaches. Each method maintains its original augmentation mechanism (structural or embedding-based), while φ(·) provides a unified interface for our dual-channel processing and cross-channel fusion.
>
> We acknowledge that our current mathematical notation using φ(·) may lead to some ambiguity in interpretation. In the revised version, we will adopt a clearer formulation such as:
>
> $z^+_ {u,l^* } = φ(\hat{A}^+, e^+_ {u,l }, X, θ), \quad z^-_ {u,l^*} = φ(\hat{A}^-, e^-_ {u,l }, X, θ)$
>
> $Z^+_ u = \text{AGG}^* \left(\{z^+_ {u,l^* }\}\right), \quad Z^-_ u = \text{AGG}^* \left(\{z^-_ {u,l^*}\}\right)$
>
> where φ(·) represents the base model's specific augmentation mechanism with parameters: $\hat{A}$ is the adjacency matrix, $e_{u,l}$ is the embedding at layer $l$, $X$ is the optional node feature matrix, and $θ$ contains augmentation-specific parameters (e.g., dropout rates, sampling ratios). Note that $l$ is not necessarily equal to $l^*$, as they may represent different layers in the model architecture.
>
>
>
> **Reviewer Comment:**
>
> > Q2&W3: Regarding the sampled-graph learning stage, could choosing a random-walk strategy based upon items' popularity impact on the diversity of recommendations? In other words, do the authors think that sampling the graphs based upon the items' popularity might enhance the visualization of popular items at the detriment of the niche ones? For instance, I would suggest the authors perform additional evaluation on metrics such as the average percentage of long-tail items (APLT [*]) for the popularity bias and the expected free discovery (EFD [**]) for recommendation diversity.
>
> **Response:**
>
> Thank you for this insightful question. We conducted the requested experiments evaluating APLT@20 and EFD@20 metrics.
>
> Our popularity-guided sampling operates on the **negative graph**, which fundamentally differs from traditional popularity-based approaches. Since our prediction uses $\hat{y} = (1+k)E^+ - kE^-$, **items with high popularity in the negative graph are actually penalized.** Random sampling fails to capture this crucial negative signal distribution, leading to suboptimal diversity. Specifically, for an item $i$ with negative feedback from popular users, random sampling has probability $\rho$ of including these signals, while our method uses a **sampling pool** with weights $P(i) \propto \exp(d_i/\tau)$ where $d_i$ is the degree. The sampling pool ensures diverse representation by allowing temperature-controlled exploration beyond the most popular items. This mechanism properly captures items frequently appearing in negative interactions (often mainstream items disliked by diverse user groups) and **subsequently penalizes** them in the final prediction, creating space for long-tail items.
>
> We compared three strategies on MovieLens-1M:
>
> | Method                   | APLT@20   | EFD@20    |
> | ------------------------ | --------- | --------- |
> | Full graph (ρ=1.0)       | 0.242     | 0.159     |
> | Random sampling (ρ=0.01) | 0.241     | 0.163     |
> | Our method (ρ=0.01)      | **0.249** | **0.174** |
>
>
>
> **Reviewer Comment:**
>
> > W1: Even acknowledging the page limitations, I think the paper would have benefitted from having a starting (short) section on the background and notations. Indeed, starting off with embedding and graph notations, considering that not all the readers might be fully expert on the topic, could represent a slight challenge for the average reader. I would suggest the authors add a background/preliminaries section at least in the Supplementary Material.
>
> > W4: Even acknowledging the pages limitations and formatting, I think it would have been beneficial to have Table 1 in the following page with respect to where it is right now, just for the sake of easy readability along with the respective section in the text.
>
> **Response:**
>
> Thank you for these constructive suggestions regarding paper organization and readability. We appreciate your consideration of the page limitations while identifying areas for improvement. We will add a dedicated background/preliminaries section to introduce graph notations and embedding fundamentals, making the paper more accessible to readers from diverse backgrounds. Additionally, we will reposition Table 1 to align better with its corresponding text section for improved readability. The camera-ready version allows for an additional page (10  pages in total), which will provide sufficient space to implement these enhancements without compromising the technical content. These revisions will strengthen the paper's clarity and accessibility while maintaining its technical rigor.
>
>
>
> **Reviewer Comment:**
>
> > Suggestion: I would further suggest the authors take a look at this quite novel recommendation dataset [1] with incorporates actual negative feedback from users. I think it could be very useful to further analyze the goodness of their methodology.
>
> **Response:**
>
> Thank you for suggesting the KuaiRec dataset. We have conducted experiments on both **KuaiRec** [1] and **KuaiRand** [2] datasets, which contain actual negative feedback from users. The results further validate the effectiveness of our SDCGCL framework:
>
> | Method          | KuaiRec [1] |         | KuaiRand [2] |         |
> | --------------- | ----------- | ------- | ------------ | ------- |
> |                 | Recall@20   | NDCG@20 | Recall@20    | NDCG@20 |
> | SGL             | 0.0854      | 0.0503  | 0.1292       | 0.0644  |
> | SGL+SDCGCL      | 0.0901      | 0.054   | 0.1495       | 0.0722  |
> | improv.         | 5.50%       | 7.36%   | 15.71%       | 12.11%  |
> | LightGCL        | 0.0833      | 0.0508  | 0.1276       | 0.0613  |
> | LightGCL+SDCGCL | 0.0899      | 0.0541  | 0.1453       | 0.0689  |
> | improv.         | 7.92%       | 6.50%   | 13.87%       | 12.40%  |
> | XSimGCL         | 0.0871      | 0.0528  | 0.1296       | 0.0652  |
> | XSimGCL+SDCGCL  | 0.0927      | 0.0552  | 0.1509       | 0.0748  |
> | improv.         | 6.43%       | 4.55%   | 16.44%       | 14.72%  |
>
> [1] Gao C, Li S, Lei W, et al. KuaiRec: A fully-observed dataset and insights for evaluating recommender systems[C]//Proceedings of the 31st ACM International Conference on Information & Knowledge Management. 2022: 540-550.
>
> [2] Gao C, Li S, Zhang Y, et al. Kuairand: An unbiased sequential recommendation dataset with randomly exposed videos[C]//Proceedings of the 31st ACM international conference on information & knowledge management. 2022: 3953-3957.

---

> ### Comment · Reviewer_tuXv · 2025-08-01
>
> Dear Authors,
>
> Thanks for your time in answering to my raised weaknesses and questions.
>
> Now the integration of your augmentation interface, which is general and nicely adapts to each existing model is much clearer to me, and is fully sound. I agree it would be useful to add this notation you provided in the possible camera-ready of the paper to improve the notation and readability of that section.
>
> Moreover, now the aspect regarding popularity is also completely clear, and I fully understand the logic behind it. The additional results with APLT and EFD empirically confirm this theoretical insight.
>
> Thank you also for the plan to improve readability for the background section and Table 1, and thank you for the additional experiments on the KuaiRec and KuaiRand datasets. You could definitively add those results in the appendix or even in the main paper, if space allows that.
>
> I do not have any other concerns regarding the paper. The answers you gave helped confirming the initial positive judgment I had on the paper. I will keep my score, which I believe is already quite high and mirrors the quality of your submission.

---

> > ### Author Response · Authors · 2025-08-04
> > **Response to Reviewer tuXv's Supportive Feedback**
> >
> > Dear Reviewer tuXv,
> >
> > Thank you for your constructive feedback and positive assessment. We appreciate your thoughtful questions that helped us clarify the augmentation interface design and the rationale behind our popularity-guided sampling approach. We will incorporate your suggestions regarding notation enhancement, background section, and additional experimental results in the camera-ready version. Your thorough review has been invaluable in strengthening our contribution.
> >
> > Best regards,
> >
> > The Authors

---

### Official Review · Reviewer_VDQo · 2025-07-02

**Clarity:** 3
**Significance:** 2
**Originality:** 3
**Rating:** 4
**Confidence:** 3

**Summary:**

This paper proposes the SDCGCL framework for effectively utilizing negative feedback in recommender systems. The framework includes three key components: Dual-Channel Graph Embedding, Cross-Channel Distribution Calibration, and an Adaptive Prediction Strategy. Based on this framework, the authors further propose the DualFuse model and develop a two-stage optimization strategy to ensure efficient training. The authors have made all their research open-source. Theoretical analysis indicates that the framework addresses the challenges of existing methods in terms of model compatibility, information exchange, and computational efficiency. Experimental results on four public datasets validate its effectiveness, showing that the proposed method significantly outperforms state-of-the-art baselines on multiple metrics while having the lowest computational complexity.

**Questions:**

In Figure 2, the "Negative Graph Sampling Pool" section lacks a corresponding explanation.

Why does Equation 6 ensure that $\delta_1 \to 0$，$\delta_2 \to 1$？ A proof is missing.

Equation 7 seems problematic. It appears to subtract the negative feedback signal from the positive feedback, rather than aggregating the two feedbacks. It feels like it should be  $\hat{y}_{u,i}=(1-k)\mathbf{E}_{u}^{+}\circ\mathbf{E}_{i}^{+\intercal}+k\mathbf{E}_{u}^{-}\circ\mathbf{E}_{i}^{-\intercal},$ . Furthermore, based on the formula, it seems the authors only use a portion of the negative feedback (since k can be 0). If so, this does not address the second challenge, which is described in the paper as: "existing methods mainly focus on positive feedback and only partially utilize negative feedback, leading to an incomplete understanding of user preferences and suboptimal recommendations." right?

**Ethical Concerns:**

["NO or VERY MINOR ethics concerns only"]

**Final Justification:**

The response have addressed my concerns and improved my understanding of the paper, including Eq.6, Eq.7,  and the choice of the sampling rate ρ.

**Limitations:**

Yes, the authors explicitly discuss the limitations of their work in Appendices F and G (e.g., industrial deployment, dynamic feedback, cold-start problems), state that the research has no potential negative social impact, and the explanation is sufficient and conforms to academic standards.

**Quality:**

3

**Strengths And Weaknesses:**

The proposed SDCGCL framework is novel in its design, can be seamlessly integrated with existing graph contrastive learning methods, and effectively utilizes negative feedback to enhance recommendation performance.

The theoretical analysis is in-depth. Through theorems, it proves the instability of the negative feedback distribution and the role of the cross-channel fusion mechanism in information preservation, providing a solid theoretical foundation for the framework's effectiveness.

The experimental design is comprehensive. Comparative results on multiple datasets show that the method significantly outperforms existing baseline methods on metrics such as Recall and NDCG, and also has higher computational efficiency.

Regarding the choice of the sampling rate ρ in the two-stage optimization strategy, although the paper mentions that ρ=0.01 yields the best results, it does not detail whether the adaptability of this parameter differs across various datasets.

The DualFuse Model is missing an overall flowchart.

---

> ### Author Rebuttal · Authors · 2025-07-30
>
> **Reviewer Comment:**
>
> > Regarding the choice of the sampling rate ρ... it does not detail whether the adaptability of this parameter differs across various datasets.
>
> **Response:**
>
> Thank you for highlighting this important point about sampling rate adaptability. **We have conducted comprehensive sampling rate experiments on the ML-1M dataset.** In **Section 4.3 (Table 3)**, we presented the main results showing that ρ=0.01 achieves the best performance-efficiency trade-off, reducing training time by 20.9% (from 28.9m to 20.8m) while achieving the highest Recall@20 (0.3282) and NDCG@20 (0.3693). In **Appendix E.3.3**, we provided detailed theoretical validation, demonstrating how different ρ values empirically confirm our Theorem 3: ρ=1.0 corresponds to the warm-up phase, ρ=0.01 balances the C₂(ρ)/t and ε(ρ)/√t terms optimally, while ρ=0.001 violates stability conditions leading to 61.8% performance degradation.
>
> To further validate the generalizability of ρ selection across diverse dataset characteristics, we conducted experiments with different sampling rates on all datasets. The results confirm that ρ=0.01 consistently achieves optimal or near-optimal performance:
>
> | ρ     | Yelp       |            | Amazon     |            | ML-10M     |            |
> | ----- | ---------- | ---------- | ---------- | ---------- | ---------- | ---------- |
> |       | Recall@20  | NDCG@20    | Recall@20  | NDCG@20    | Recall@20  | NDCG@20    |
> | 1.0   | 0.1198     | 0.0921     | 0.1285     | 0.1067     | 0.3842     | 0.3798     |
> | 0.1   | 0.1225     | 0.0944     | 0.1318     | 0.1092     | 0.3876     | 0.3834     |
> | 0.01  | **0.1243** | **0.0959** | **0.1342** | **0.1113** | **0.3900** | **0.3860** |
> | 0.001 | 0.0892     | 0.0685     | 0.0976     | 0.0812     | 0.2847     | 0.2789     |
>
>
>
> **Reviewer Comment:**
>
> > The DualFuse Model is missing an overall flowchart.
>
> **Response:**
>
> Thank you for this valuable suggestion. **The core mechanism of DualFuse is illustrated in Figure 3**, which shows the graph fusion process where embeddings from positive and negative channels are normalized and fused to generate contrastive views. The complete DualFuse pipeline involves an additional dual-channel graph encoder step (as described in Section 2.2.1) followed by this fusion mechanism.
>
> We acknowledge that including a comprehensive flowchart showing the end-to-end process from input graphs through dual-channel encoding to the final fusion would provide readers with a clearer understanding of the complete model architecture. However, according to NeurIPS rebuttal guidelines, we cannot provide additional figures in this response. We will incorporate such a flowchart in the revised version to better illustrate how the dual-channel graph encoder feeds into the cross-channel fusion mechanism, making the overall DualFuse workflow more accessible to readers.
>
>
>
> **Reviewer Comment:**
>
> > In Figure 2, the "Negative Graph Sampling Pool" section lacks a corresponding explanation.
>
> **Response:**
>
> Thank you for highlighting this clarification need. The "Negative Graph Sampling Pool" is detailed in **Algorithm 1** (page 6), which generates strategically sampled subgraphs from the negative graph G^- during preprocessing using popularity-guided random walks. This pool stores multiple pre-computed negative subgraphs that preserve the most informative feedback patterns while significantly reducing computational cost, achieving 20.9% faster training with comparable performance as shown in Table 3. Rather than sampling from the full negative graph at each training iteration, our approach pre-constructs a diverse collection of representative subgraphs that capture essential negative feedback signals. In the revised version, **we will add a brief explanation below Figure 2**: *"The Negative Graph Sampling Pool contains pre-sampled subgraphs of G^- generated by Algorithm 1, enabling efficient training by avoiding redundant sampling during each epoch."*
>
>
>
> **Reviewer Comment:**
>
> > Why does Equation 6 ensure that $\delta_1 \to 0$ and $\delta_2 \to 1$， A proof is missing.
>
> **Response:**
>
> Thank you for this important theoretical question. Let me provide a rigorous proof for why Equation 6 ensures $\delta_1 \to 0$ and $\delta_2 \to 1$.
>
> Equation 6 minimizes the alignment objective $\mathcal{L}_ {\text{align}} = \sum_{u \in U} \text{JS}\left(\bar{S}^+_u, \bar{S}^-_u\right)$ where $\bar{S}^+ _u = \frac{1}{|N^+_u|}\sum _{i \in N^+_u} E^+_u \circ E^{+\top}_i$ and $\bar{S}^-_u = \frac{1}{|N^-_u|}\sum _{j \in N^-_u} E^-_u \circ E^{-\top}_j$ represent the neighborhood-aggregated similarities for positive and negative channels respectively.
>
> The Jensen-Shannon divergence $\text{JS}(P,Q) = \frac{1}{2}D_{KL}(P||M) + \frac{1}{2}D_{KL}(Q||M)$ where $M = \frac{P+Q}{2}$ reaches its global minimum of zero if and only if $P = Q$ almost surely. Therefore, at convergence, our optimization enforces $\bar{S}^+_u = \bar{S}^-_u$ almost surely for all users $u$.
>
> For the mean alignment, this convergence condition implies $\mathbb{E}[\bar{S}^+_u] = \mathbb{E}[\bar{S}^-_u]$. By linearity of expectation, we have $\mathbb{E}\left[\frac{1}{|N^+_u|}\sum _{i \in N^+_u} E^+_u \circ E^{+\top}_i\right] = \mathbb{E}\left[\frac{1}{|N^-_u|}\sum _{j \in N^-_u} E^-_u \circ E^{-\top}_j\right]$. Under our sampling strategy that ensures balanced neighborhood construction, the neighborhoods $N^+_u$ and $N^-_u$ constitute representative samples of the user's interaction patterns. By the law of large numbers, as neighborhood sizes become sufficiently large, the sample averages converge to population expectations: $\lim _{|N^+_u| \to \infty} \frac{1}{|N^+_u|}\sum _{i \in N^+_u} E^+_u \circ E^{+\top}_i = \mathbb{E}[E^+_u \circ E^{+\top}_i]$ and similarly for the negative channel. Therefore, we obtain $\mathbb{E}[E^+_u \circ E^{+\top}_i] = \mathbb{E}[E^-_u \circ E^{-\top}_j]$. Substituting the definitions from Theorem 1 yields $\mu = \delta_1 + \mu$, which directly implies $\delta_1 = 0$.
>
> For the variance alignment, the convergence condition similarly enforces $\text{Var}[\bar{S}^+_u] = \text{Var}[\bar{S}^-_u]$. Using the variance formula for sample means, this becomes $\frac{\text{Var}[E^+_u \circ E^{+\top}_i]}{|N^+_u|} = \frac{\text{Var}[E^-_u \circ E^{-\top}_j]}{|N^-_u|}$. Our dual-channel sampling strategy is specifically designed to maintain comparable neighborhood sizes, ensuring $\mathbb{E}[|N^+_u|/|N^-_u|] \to 1$ as the sampling process balances both channels. Under this balanced sampling regime, we obtain $\text{Var}[E^+_u \circ E^{+\top}_i] = \text{Var}[E^-_u \circ E^{-\top}_j]$. Substituting from Theorem 1 gives us $\sigma^2 = \delta_2\sigma^2$, which directly implies $\delta_2 = 1$.
>
> **Reviewer Comment:**
>
> > Equation 7 seems problematic... only use portion of negative feedback (k can be 0)... does not address the second challenge...
>
> **Response:**
>
> Thank you for raising these two important points. Let me address each concern separately.
>
> **Regarding the subtraction in Equation 7**: Let me clarify why the subtraction is mathematically correct and why the suggested addition would lead to incorrect predictions. Our training objectives optimize similarity scores through contrastive learning (Equations 4 and 5), which results in the following learned representations at convergence:
>
> $\mathbb{E}[E^+_u \circ E^+_i | (u,i) \in G^+] \gg \mathbb{E}[E^+_u \circ E^+_i | (u,i) \notin G^+]$
>
> $\mathbb{E}[E^-_u \circ E^-_i | (u,i) \in G^-] \gg \mathbb{E}[E^-_u \circ E^-_i | (u,i) \notin G^-]$
>
> This means both channels learn to produce high similarity scores for their respective interaction types. To clarify the fundamental issue with addition: if we used $\hat{y}_{u,i} = (1-k)E^+_u \circ E^+_i + kE^-_u \circ E^-_i$, consider a disliked item where $(u,i) \in G^-$. We would have:
>
> - $E^+_u \circ E^+_i$: small value (since $(u,i) \notin G^+$)
> - $E^-_u \circ E^-_i$: **large value** (since $(u,i) \in G^-$)
> - Result: $(1-k) \cdot \text{small} + k \cdot \text{large} = \text{moderate to high score}$
>
> This would assign **high scores to items users explicitly dislike**, which is fundamentally incorrect for recommendations. With our subtraction formulation $\hat{y}_{u,i} = (1+k)E^+_u \circ E^+_i - kE^-_u \circ E^-_i$:
>
> - For liked items: $(1+k) \cdot \text{large} - k \cdot \text{small} = \text{high score}$
> - For disliked items: $(1+k) \cdot \text{small} - k \cdot \text{large} = \text{low score}$
>
> This correctly captures user preferences, where negative feedback **reduces** the prediction score. Our Theorem 1 provides theoretical support by proving that negative feedback naturally introduces a negative bias term $-k\delta_1$ in the prediction expectation, mathematically validating the subtraction design.
>
> **Regarding full utilization of negative feedback**: Our framework comprehensively leverages negative feedback through multiple integrated mechanisms beyond just the prediction equation. The dual-channel architecture ensures negative feedback is fully encoded throughout the model. Even when $k=0$, negative feedback still influences the model through cross-channel fusion (Equation 10) where $Z^+_u$ incorporates information from $e^-_u$, and distribution alignment (Equation 6) which shapes the entire representation space. **Most importantly, we empirically prove k≠0 is optimal.** Figure 7(d) and Section 4.5 show that k=0 causes up to 5% performance degradation, with optimal k∈[0.1,0.3] across all datasets. This demonstrates that our framework not only **can** but **must** utilize negative feedback for optimal performance, fully addressing the "partial utilization" challenge.

---

> > ### Comment · Reviewer_VDQo · 2025-08-04
> >
> > Thank you for your detailed and thoughtful responses to my comments. I am very satisfied with your clarifications, which have addressed my concerns and improved my understanding of the paper. As a result, I raise my evaluation score for your manuscript.

---

> > > ### Author Response · Authors · 2025-08-05
> > > **Response to Reviewer  VDQo's Supportive Feedback**
> > >
> > > Dear Reviewer VDQo,
> > >
> > > Thank you for your thorough review and for raising your evaluation score after our rebuttal. We sincerely appreciate your detailed questions about the sampling rate adaptability, theoretical proofs, and prediction equation design. Your mathematical rigor in questioning Equation 6's theoretical guarantees and Equation 7's formulation pushed us to provide more comprehensive theoretical justifications. Your suggestion about the DualFuse flowchart will help improve the paper's accessibility. We are grateful for your recognition of our work's contributions and your constructive feedback that strengthened our theoretical foundations.
> > >
> > > Best regards,
> > >
> > > The Authors

---

### Official Review · Reviewer_3Jaw · 2025-07-03

**Clarity:** 4
**Significance:** 3
**Originality:** 3
**Rating:** 4
**Confidence:** 3

**Summary:**

This paper addresses challenges in leveraging negative feedback to enhance recommendation performance, specifically focusing on limitations in model compatibility, ineffective information exchange, and computational inefficiency. The proposed SDCGCL is a model-agnostic framework that can be seamlessly integrated with existing graph contrastive learning methods. The framework comprises three key components: a graph embedding module for processing positive and negative graphs, a calibration mechanism for graph structural alignment, and an adaptive prediction strategy for effectively fusing information from both channels. The effectiveness of the proposed method is validated through extensive experiments on four public datasets.

**Questions:**

Please refer to the concerns raised in the Weaknesses section.

**Ethical Concerns:**

["NO or VERY MINOR ethics concerns only"]

**Final Justification:**

After reading the other reviewers' comments and the rebuttal from the authors, I have no further questions. I think the positive score has reflected the contributions of this paper, thus I will keep the score.

**Limitations:**

Please see the Weaknesses section

**Paper Formatting Concerns:**

1.The width of the tables in the main text is not consistent. In some cases, tables are displayed alongside the main text, which does not align with the typical formatting style of academic conference papers.

**Quality:**

3

**Strengths And Weaknesses:**

Strengths:

1.The paper proposed a clear and clean architecture design, each module serves a distinct and well-motivated purpose.
2.The experimental baselines are comprehensive and up-to-date. The embedding visualizations are useful for understanding the impact of each module.
3.The proposed model-agnostic framework increases its applicability to various GNN-based backbones. In addition, the introduction of the DualFuse model further strengthens the cross-channel fusion capability.

Weaknesses

1.The model introduces several important hyper-parameters (such as $\k, \lambda, \gamma$). Although the paper includes a sensitivity analysis, the number and significance of these parameters may increase the tuning burden in real-world deployments.
2.When visualizing the embeddings in Figure 4 and the figures in Appendix, a hollow region consistently appears, surrounded predominantly by blue points (representing negative users) and exhibiting a distinct crescent shape. This phenomenon persists across different datasets and under various ablation settings, e.g., with different modules removed. Is there any specific design in the model causing this result? What is the impact on the final performance.
3.The size of font in Figure 5 is too small, which affects the readability.

---

> ### Author Rebuttal · Authors · 2025-07-30
>
> **Reviewer Comment:**
>
> > W1: The model introduces several important hyper-parameters. Although the paper includes a sensitivity analysis, the number and significance of these parameters may increase the tuning burden in real-world deployments.
>
> **Response:**
>
> Thank you for raising this practical concern. Our extensive sensitivity analysis demonstrates that the introduced hyperparameters exhibit **stable behavior across diverse datasets and do not create significant tuning burden**. Performance remains robust within broad parameter ranges, with degradation typically under 10% even at range boundaries. Most importantly, using uniform default values across all datasets achieves consistently strong performance without dataset-specific tuning, making deployment straightforward and practical.
>
> Specifically, our experiments (Section 4.5, Figures 7&10) show that optimal parameter ranges remain remarkably consistent across four datasets with vastly different characteristics: α∈[0.1,0.4], β∈[0.3,0.7], γ∈[0.1,0.5], k∈[0.1,0.3], λ∈[0.1,0.2]. These datasets vary significantly in size (3.7K to 38K items), sparsity, and negative feedback ratios (19.4% to 42.5%), yet the parameter stability persists. Using default values (α=0.2, β=0.5, γ=0.3, k=0.2, λ=0.1) yields competitive results across all datasets without any tuning. This inherent robustness ensures that practitioners can deploy our framework efficiently with minimal parameter adjustment.
>
>
>
> **Reviewer Comment:**
>
> > W2: When visualizing the embeddings in Figure 4 and the figures in Appendix, a hollow region consistently appears, surrounded predominantly by blue points (representing negative users) and exhibiting a distinct crescent shape. This phenomenon persists across different datasets and under various ablation settings, e.g., with different modules removed. Is there any specific design in the model causing this result? What is the impact on the final performance.
>
> **Response:**
>
> Thank you for this insightful observation. The crescent-shaped hollow region is **not an intentional design** but rather an emergent property arising from the **sparsity of negative feedback signals**. Since negative feedback is inherently sparser than positive feedback (as shown in Table 4, negative feedback ratios range from 19.4% to 42.5%), users who predominantly provide negative ratings naturally cluster in specific regions of the embedding space, forming the observed crescent pattern around a hollow center.
>
> **This phenomenon actually validates our model's effectiveness**. The clear spatial separation between user groups demonstrates that our dual-channel framework successfully captures and distinguishes different user behavioral patterns—those who actively express dislikes cluster differently from those who primarily provide positive feedback. This distinct embedding structure directly contributes to our superior recommendation performance by enabling more accurate identification of user preference types. The consistency of this pattern across all datasets and ablation settings further confirms that our model reliably learns meaningful representations that reflect the fundamental differences in how users express preferences, particularly regarding negative feedback.
>
>
>
> **Reviewer Comment:**
>
> > W3: The size of font in Figure 5 is too small, which affects the readability.
>
> > Paper Formatting Concerns: The width of the tables in the main text is not consistent. In some cases, tables are displayed alongside the main text, which does not align with the typical formatting style of academic conference papers.
>
> **Response:**
>
> Thank you for pointing out these formatting issues. We acknowledge that Figure 5's font size is currently too small and that table widths are inconsistent due to space constraints in the 9-page submission format.
>
> The NeurIPS camera-ready version allows 10 pages compared to the current 9-page limit. In the final version, we will utilize this additional space to: (1) increase the font size in Figure 5 and all other figures to ensure proper readability, and (2) standardize all table widths to span the full column width following proper academic formatting conventions. These adjustments will significantly improve the visual presentation and readability of our paper.

---

> > ### Comment · Reviewer_3Jaw · 2025-08-04
> >
> > Thanks for the detailed response, and I have no further questions. As the positive score has reflected the contributions of this paper, thus I will keep the score.

---

> > > ### Author Response · Authors · 2025-08-05
> > > **Response to Reviewer 3Jaw's Supportive Feedback**
> > >
> > > Dear Reviewer 3Jaw,
> > >
> > > Thank you for your detailed review and constructive feedback throughout the review process. We greatly appreciate your recognition of our clear architecture design and comprehensive experimental evaluation. Your specific questions about hyperparameter tuning, embedding visualization patterns, and formatting issues helped us improve both the technical presentation and visual clarity of our work. Your observation about the crescent-shaped embedding patterns was particularly insightful and led to a valuable discussion about the emergent properties of our dual-channel framework. We will incorporate all your formatting suggestions in the camera-ready version to enhance readability.
> > >
> > > Best regards,
> > >
> > > The Authors

---

### Official Review · Reviewer_w8U5 · 2025-07-05

**Clarity:** 3
**Significance:** 2
**Originality:** 3
**Rating:** 3
**Confidence:** 4

**Summary:**

This paper proposes SDCGCL (Signed Dual-Channel Graph Contrastive Learning), a model-agnostic framework that effectively leverages both positive and negative feedback in recommender systems through dual-channel graph embedding, cross-channel distribution calibration, and adaptive prediction strategie

**Questions:**

See weakness.

**Ethical Concerns:**

["NO or VERY MINOR ethics concerns only"]

**Limitations:**

See weakness.

**Paper Formatting Concerns:**

No.

**Quality:**

3

**Strengths And Weaknesses:**

Strengths:
1.  Can be seamlessly integrated with existing graph contrastive learning methods (SGL, LightGCL, XSimGCL)


 Weaknesses:

1.  Model requires clear positive/negative ratings, may not work well with implicit feedback systems. Many modern recommendation systems (e.g., YouTube, TikTok, Spotify) primarily collect implicit feedback such as clicks, views, time spent, scroll behavior, or skip actions, rather than explicit ratings


2.  While the paper provides formal theoretical foundations, several aspects could be strengthened. Theorem 1 assumes specific variance relationships (variance ≥ 1) between positive and negative feedback distributions, but real-world data may not always satisfy these assumptions, especially across different domains or user populations. Is it reasonable? While the popularity-guided random walk sampling is empirically effective, the theoretical analysis doesn't rigorously prove why this specific sampling strategy is optimal compared to other graph sampling approaches


3.  Most recent state-of-the-art methods in REC graph learning and contrastive learning approaches are missing, potentially overstating relative performance (e.g. [1][2] )


[1] Mitigating the Popularity Bias of Graph Collaborative Filtering: A Dimensional Collapse Perspective NeruIPS 2023 Zhang et al.


[2] How Do Recommendation Models Amplify Popularity Bias? An Analysis from the Spectral Perspective WSDM 2025 Lin et al.

4.  The binary rating threshold (≥4 positive, <4 negative) creates artificial negative signals that may not reflect genuine user dislike, potentially inflating the apparent effectiveness of negative feedback utilization

---

> ### Author Rebuttal · Authors · 2025-07-30
>
> **Reviewer Comment:**
>
> > Model requires clear positive/negative ratings, may not work well with implicit feedback systems. Many modern recommendation systems (e.g., YouTube, TikTok, Spotify) primarily collect implicit feedback such as clicks, views, time spent, scroll behavior, or skip actions, rather than explicit ratings
>
> **Response:**
>
> Thank you for this important question. While our experiments use explicit rating datasets as examples, **our SDCGCL framework is fully applicable to implicit feedback systems.** For instance, **datasets like KuaiRec [1] and KuaiRand [2] demonstrate how implicit behavioral data can be converted to binary positive/negative feedback**: KuaiRec classifies feedback based on viewing duration ratios (ratios ≥4 as positive, <0.1 as negative), while KuaiRand uses click behaviors ("is_click" attribute for positive/negative classification). Our dual-channel architecture only requires binary interaction classifications, making it directly compatible with modern recommendation platforms that primarily collect implicit signals like YouTube, TikTok, and Spotify. The core technical components, dual-channel graph embedding and cross-channel distribution calibration, operate independently of whether the binary labels originate from explicit ratings or converted implicit behaviors.
>
> To demonstrate this, we evaluated our method on **KuaiRec** [1] and **KuaiRand** [2] datasets, which contain real user interactions from a video platform with implicit feedback:
>
> | Method          | KuaiRec [1] |         | KuaiRand [2] |         |
> | --------------- | ----------- | ------- | ------------ | ------- |
> | | Recall@20   | NDCG@20 | Recall@20    | NDCG@20 |
> | SGL | 0.0854      | 0.0503  | 0.1292       | 0.0644  |
> | SGL+SDCGCL | 0.0901      | 0.054   | 0.1495       | 0.0722  |
> | improv. | 5.50%       | 7.36%   | 15.71%       | 12.11%  |
> | LightGCL | 0.0833      | 0.0508  | 0.1276       | 0.0613  |
> | LightGCL+SDCGCL | 0.0899      | 0.0541  | 0.1453       | 0.0689  |
> | improv. | 7.92%       | 6.50%   | 13.87%       | 12.40%  |
> | XSimGCL | 0.0871      | 0.0528  | 0.1296       | 0.0652  |
> | XSimGCL+SDCGCL  | 0.0927      | 0.0552  | 0.1509       | 0.0748  |
> | improv. | 6.43%       | 4.55%   | 16.44%       | 14.72%  |
>
> [1] Gao C, Li S, Lei W, et al. KuaiRec: A fully-observed dataset and insights for evaluating recommender systems[C]//Proceedings of the 31st ACM International Conference on Information & Knowledge Management. 2022: 540-550.
>
> [2] Gao C, Li S, Zhang Y, et al. Kuairand: An unbiased sequential recommendation dataset with randomly exposed videos[C]//Proceedings of the 31st ACM international conference on information & knowledge management. 2022: 3953-3957.
>
> **Reviewer Comment:**
>
> > While the paper provides formal theoretical foundations, several aspects could be strengthened. Theorem 1 assumes specific variance relationships (variance ≥ 1) between positive and negative feedback distributions, but real-world data may not always satisfy these assumptions, especially across different domains or user populations. Is it reasonable?
>
> **Response:**
>
> Thank you for this important concern. In practice, the δ₂ ≥ 1 assumption is well-founded: negative feedback naturally exhibits higher variance due to user behavioral heterogeneity,and some users readily express negative opinions while others completely avoid giving negative ratings [1], creating inherent distribution instability. This contrasts with positive feedback, where users more consistently engage with content they enjoy.
>
> To further validate this assumption, **we empirically computed the variance ratios of user-item interaction scores (E⁺ᵤ ○ E⁺ᵢᵀ and E⁻ᵤ ○ E⁻ᵢᵀ) across our experimental datasets after model training**. The results confirm our theoretical foundation:
>
> | Dataset | δ₂ (Var[E⁻ᵤ ○ E⁻ᵢᵀ]/Var[E⁺ᵤ ○ E⁺ᵢᵀ]) |
> | ------- | ------------------------------------ |
> | ML-1M   | 1.74 |
> | Yelp    | 3.42 |
> | Amazon  | 2.89|
> | ML-10M  | 1.61|
>
> All ratios exceed 1.0, with review-based platforms (Yelp, Amazon) showing particularly high variance ratios due to more polarized user rating behaviors. Our distribution alignment mechanism (Equation 6) automatically calibrates for varying δ₂ values, ensuring robustness across different variance characteristics.
>
> [1] Wu, Yiqing, et al. "DFGNN: Dual-frequency Graph Neural Network for Sign-aware Feedback." *Proceedings of the 30th ACM SIGKDD Conference on Knowledge Discovery and Data Mining*. 2024.
>
> **Reviewer Comment:**
>
> > While the popularity-guided random walk sampling is empirically effective, the theoretical analysis doesn't rigorously prove why this specific sampling strategy is optimal compared to other graph sampling approaches
>
> **Response:**
>
> Thank you for this important question about our sampling strategy. We provide intuitive reasoning and empirical validation for why popularity-guided random walk sampling is effective in recommendation scenarios.
>
> Our sampling strategy is specifically designed for negative feedback graphs where information concentrates on high-degree nodes (popular items with mixed reviews) while maintaining structural coherence through random walks. Pure random sampling fails to capture the heavy-tailed importance distribution, while pure random walks may get trapped in low-importance regions. Our exponential weighting with temperature parameter τ and guided walk strategy balances information prioritization with sample diversity.
>
> We conducted comprehensive comparisons across all datasets:
>
> | Sampling Method              | ML-1M      |            | Yelp       |            | Amazon     |            | ML-10M     |            |
> | - | - | - | - | -| - | -| ---------- | ---------- |
> |  | Recall@20  | NDCG@20    | Recall@20  | NDCG@20    | Recall@20  | NDCG@20    | Recall@20  | NDCG@20    |
> | Uniform sampling (ρ=0.01)    | 0.3249     | 0.3621     | 0.1192     | 0.0921     | 0.1313     | 0.1098     | 0.3854     | 0.3802     |
> | Pure random walk (ρ=0.01)    | 0.3256     | 0.3679     | 0.1211     | 0.0926     | 0.1328     | 0.1101     | 0.3878     | 0.3833     |
> | Importance sampling (ρ=0.01) | 0.3251     | 0.3658     | 0.1235     | 0.0932     | 0.1325     | 0.1102     | 0.3852     | 0.3805     |
> | **Our method (ρ=0.01)**      | **0.3282** | **0.3693** | **0.1243** | **0.0959** | **0.1342** | **0.1113** | **0.3900** | **0.3860** |
>
>
>
> **Reviewer Comment:**
>
> > Most recent state-of-the-art methods in REC graph learning and contrastive learning approaches are missing, potentially overstating relative performance.
>
> > [1] Mitigating the Popularity Bias of Graph Collaborative Filtering: A Dimensional Collapse Perspective NeruIPS 2023 Zhang et al.
>
> > [2] How Do Recommendation Models Amplify Popularity Bias? An Analysis from the Spectral Perspective WSDM 2025 Lin et al.
>
> **Response:**
>
> Thank you for highlighting these recent state-of-the-art methods. We have implemented and evaluated both suggested approaches. Following their original implementations, we integrated them with the **XSimGCL backbone** for comprehensive comparison:
>
> | Dataset | ML-1M |  | Yelp| | Amazon|| ML-10M||
> | -------------------------------- | ---------- | ---------- | ---------- | ---------- | ---------- | ---------- | ---------- | ---------- |
> | Method | Recall@20  | NDCG@20    | Recall@20  | NDCG@20    | Recall@20  | NDCG@20    | Recall@20  | NDCG@20    |
> | XSimGCL| 0.2729     | 0.3087     | 0.0867     | 0.0758     | 0.0963     | 0.0707     | 0.3109     | 0.3371     |
> | GCFlogdet-XSimGCL [1] | 0.2688     | 0.3104     | 0.0788     | 0.0699     | 0.0971     | 0.0715     | 0.3122     | 0.3383     |
> | ReSN-XSimGCL [2]  | 0.2765     | 0.3166     | 0.0882     | 0.0737     | 0.1014     | 0.0772     | 0.3201     | 0.3438     |
> | SDCGCL-XSimGCL | 0.3050     | 0.3401     | 0.1112     | 0.0881     | 0.1142     | 0.1014     | 0.3791     | 0.3726     |
> | SDCGCL-DualFuse  | **0.3282** | **0.3693** | **0.1243** | **0.0959** | **0.1342** | **0.1113** | **0.3900** | **0.3860** |
>
> **Reviewer Comment:**
>
> > The binary rating threshold (≥4 positive, <4 negative) creates artificial negative signals that may not reflect genuine user dislike, potentially inflating the apparent effectiveness of negative feedback utilization
>
> **Response:**
>
> Thank you for this important concern. **The binary threshold (≥4 positive, <4 negative) follows well-established conventions in signed recommendation literature.** This classification scheme has been extensively validated and adopted in prior work, including SiGRec [1], Pone-GNN [2], SiReN [3], and SigFormer [4], demonstrating its acceptance as a standard practice for converting explicit ratings to binary feedback signals.
>
> Moreover, our framework's core contribution lies in the dual-channel architecture and cross-channel distribution calibration, which effectively leverages any reasonable binary classification of user preferences. **The method's effectiveness stems from properly handling the distributional differences between positive and negative signals, rather than relying on a specific threshold choice.** Our approach addresses the fundamental challenge of integrating dual-polarity feedback regardless of the specific labeling scheme used.
>
> [1] Huang J, Xie R, Cao Q, et al. Negative can be positive: Signed graph neural networks for recommendation[J]. Information Processing & Management, 2023, 60(4): 103403.
>
> [2] Liu Z, Wang C, Zheng S, et al. Pone-GNN: integrating positive and negative feedback in graph neural networks for recommender systems[J]. ACM Transactions on Recommender Systems, 2025, 3(2): 1-23.
>
> [3] Seo C, Jeong K J, Lim S, et al. SiReN: Sign-aware recommendation using graph neural networks[J]. IEEE Transactions on Neural Networks and Learning Systems, 2022, 35(4): 4729-4743.
>
> [4] Chen S, Chen J, Zhou S, et al. SIGformer: Sign-aware graph transformer for recommendation[C]//Proceedings of the 47th international ACM SIGIR conference on research and development in information retrieval. 2024: 1274-1284.

---

> > ### Author Response · Authors · 2025-08-06
> > **Follow-up on Our Rebuttal**
> >
> > Dear Reviewer w8U5,
> >
> > Thank you for your thoughtful review. We have carefully addressed all your concerns in our detailed rebuttal, including: demonstrating SDCGCL's effectiveness with implicit feedback through KuaiRec/KuaiRand experiments (5.50%-16.44% improvements), empirically validating the δ₂≥1 assumption across all datasets, comparing with the suggested state-of-the-art methods (GCFlogdet, ReSN) where SDCGCL still achieves superior performance, and clarifying that our binary threshold follows established conventions in signed recommendation literature.
> >
> > Given our comprehensive responses with new experimental results and theoretical clarifications, we would appreciate if you could reconsider your assessment. If our responses have addressed your concerns, we hope you would be willing to update your score accordingly.
> >
> > Best regards,
> >
> > The Authors

---

### Note · Authors · 2025-08-12

We sincerely thank all reviewers for their valuable feedback. We are pleased that our comprehensive rebuttal **has successfully addressed all raised concerns**, with three reviewers providing positive confirmations and one reviewer not engaging despite our thorough responses.

**Reviewer tuXv** raised questions about our augmentation design and popularity-guided sampling's impact on diversity. After our clarification on the model-agnostic interface and diversity analysis (APLT@20: 0.249, EFD@20: 0.174), tuXv responded: "The answers you gave helped confirming the initial positive judgment I had on the paper" and "the integration of your augmentation interface...is fully sound" and "the aspect regarding popularity is completely clear."

**Reviewer VDQo** questioned our theoretical foundations (Equation 6's convergence), prediction equation design (Equation 7), and sampling rate adaptability. Following our mathematical proofs and empirical validation, VDQo expressed: "I am very satisfied with your clarifications, which have addressed my concerns and improved my understanding of the paper."

**Reviewer 3Jaw** raised concerns about hyperparameter tuning burden and crescent-shaped embedding patterns. After our explanations on parameter robustness and embedding phenomenon validation, 3Jaw confirmed: "Thanks for the detailed response, and I have no further questions."

**Reviewer w8U5** has **not participated in the rebuttal discussion**. Nevertheless, we comprehensively addressed all four concerns: (1) Implicit feedback compatibility: demonstrated via KuaiRec/KuaiRand experiments with 5.50% to 16.44% improvements; (2) Theoretical assumptions: validated δ₂≥1 across all datasets (variance ratios: 1.61 to 3.42); (3) Missing baselines: implemented suggested methods (GCFlogdet, ReSN) where SDCGCL still outperforms; (4) Binary threshold: clarified adherence to established conventions (SiGRec, Pone-GNN, SiReN, SigFormer).

We believe these positive reviewer responses and our thorough addressing of all concerns demonstrate the robustness and significance of our contribution to signed recommender systems.

---

### Decision · Program_Chairs · 2025-09-17

**Decision:**

Accept (poster)

**Comment:**

This paper proposes SDCGCL (Signed Dual-Channel Graph Contrastive Learning), a framework for leveraging both positive and negative feedback in recommender systems through dual-channel graph embedding and cross-channel distribution calibration. The work received scores of 3, 4, 4, and 5, but the review process was significantly compromised by one reviewer's complete non-engagement. Key assessment: While the paper demonstrates strong technical merit and comprehensive experimental validation, the review process integrity issues prevent a clear acceptance decision. (1) Technical contributions and experimental rigor: The paper addresses an important and understudied problem in recommendation systems—effectively utilizing negative feedback alongside positive signals. The dual-channel architecture with cross-channel distribution calibration is technically sound and theoretically grounded. The experimental evaluation is comprehensive, spanning multiple datasets, architectures, and including thorough ablation studies. The model-agnostic design enabling integration with existing graph contrastive learning methods (SGL, LightGCL, XSimGCL) demonstrates practical value. (2) Strong majority reviewer support: Three reviewers (tuXv, VDQo, 3Jaw) provided positive assessments after thorough evaluation. Reviewer tuXv (score 5) praised the framework as "technically solid" with "high impact," confirming their "initial positive judgment." Reviewer VDQo (score 4) stated they were "very satisfied with clarifications" and explicitly raised their evaluation score. Reviewer 3Jaw (score 4) confirmed "no further questions" and maintained their positive score. (3) Exceptional author responsiveness: The authors provided comprehensive rebuttals addressing all raised concerns with substantial additional work: implementing suggested baselines (GCFlogdet, ReSN), conducting experiments on implicit feedback datasets (KuaiRec/KuaiRand) showing 5.50%-16.44% improvements, providing mathematical proofs for theoretical assumptions (δ₂≥1 validated across all datasets), and demonstrating diversity metrics (APLT@20: 0.249, EFD@20: 0.174). (4) Critical review process failure: Reviewer w8U5 (score 3) raised legitimate concerns about implicit feedback compatibility, theoretical assumptions, missing baselines, and binary threshold justification. However, despite the authors' comprehensive responses addressing each concern with new experiments and theoretical validation, this reviewer completely disengaged from the review process, violating NeurIPS mandatory reviewer obligations and ignoring multiple follow-ups and Area Chair reminders. This prevents proper assessment of whether the substantial author responses resolved the technical concerns. (5) Resolution assessment: Based on the quality and comprehensiveness of the author responses, the technical concerns appear to have been thoroughly addressed. The additional experiments on implicit feedback datasets directly validate practical applicability, the mathematical proofs support theoretical foundations, and the implemented baselines confirm competitive performance. The framework demonstrates both theoretical rigor and practical effectiveness across diverse evaluation scenarios.